# Test-Time Adaptation of Vision-Language Models for Open-Vocabulary Semantic Segmentation

Mehrdad Noori*      David Osowiechi*      Gustavo A. Vargas Hakim      Ali Bahri

Moslem Yazdanpanah      Sahar Dastani      Farzad Beizaee      Ismail Ben Ayed

Christian Desrosiers

LIVIA, ÉTS Montréal, Canada
International Laboratory on Learning Systems (ILLS)

## Abstract

Recently, test-time adaptation has attracted wide interest in the context of vision-language models for image classification. However, to the best of our knowledge, the problem is completely overlooked in dense prediction tasks such as Open-Vocabulary Semantic Segmentation (OVSS). In response, we propose a novel TTA method tailored to adapting VLMs for segmentation during test time. Unlike TTA methods for image classification, our Multi-Level and Multi-Prompt (MLMP) entropy minimization integrates features from intermediate vision-encoder layers and is performed with different text-prompt templates at both the global CLS token and local pixel-wise levels. Our approach could be used as plug-and-play for any segmentation network, does not require additional training data or labels, and remains effective even with a single test sample. Furthermore, we introduce a comprehensive OVSS TTA benchmark suite, which integrates a rigorous evaluation protocol, nine segmentation datasets, 15 common synthetic corruptions, and additional real and rendered domain shifts, **with a total of 87 distinct test scenarios**, establishing a standardized and comprehensive testbed for future TTA research in open-vocabulary segmentation. Our experiments on this suite demonstrate that our segmentation-tailored method consistently delivers significant gains over direct adoption of TTA classification baselines. Code and data are available at `https://github.com/dosowiechi/MLMP`.

## 1 Introduction

Contrastive Vision-Language Models (VLMs) such as CLIP [1] have demonstrated remarkable generalization capabilities by aligning vision and language modalities through large-scale pre-training. This versatility has positioned VLMs as powerful foundation models for numerous downstream tasks [2, 3, 4]. A promising direction for leveraging VLMs beyond classification is Open-Vocabulary Semantic Segmentation (OVSS), where models aim to segment objects beyond a pre-defined set of categories, via VLMs' zero-shot recognition capabilities. Unlike traditional segmentation methods that require pixel-wise supervision, OVSS enables generalization to unseen object categories through language-driven representations.

Although existing OVSS methods have made significant progress, they remain vulnerable to domain shifts at test time, such as environmental changes or image corruptions, which may dramatically degrade segmentation quality. In the absence of a mechanism enabling them to adapt to unseen test-time distributions, these models might lose their generalization capabilities, which limits their reliability in real-world applications. Consequently, there is an unresolved gap for the Test-Time

---

*Equal contribution.
Correspondence to mehrdad.noori.1@ens.etsmtl.ca and david.osowiechi.1@ens.etsmtl.ca

39th Conference on Neural Information Processing Systems (NeurIPS 2025).

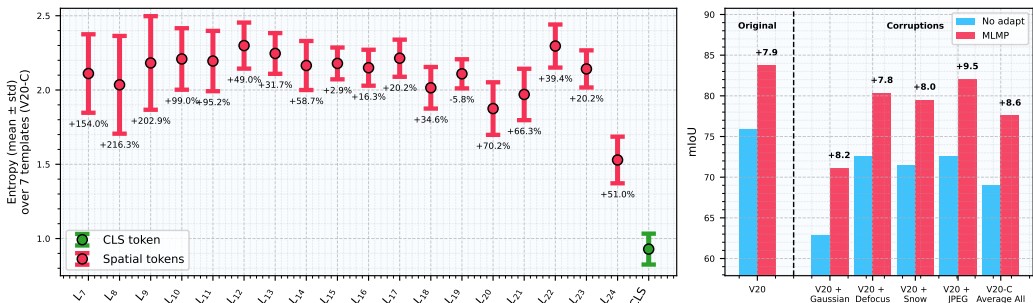

Figure 1: **Motivation. (a) Left**: Mean $\pm$ std entropy across seven text templates for the CLS token and the spatial tokens of the final and intermediate vision layers. Even the final-layer spatial tokens exhibit higher entropy and variability than CLS, and this sensitivity grows further in intermediate layers (numbers show % std increase relative to CLS). These patterns highlight pronounced prompt-induced uncertainty at multiple depths and motivate both multi-level and multi-prompt adaptation. **(b) Right**: mIoU of the baseline vs. MLMP on clean and corrupted data, showing consistent absolute improvements and underscoring the effectiveness of our joint adaptation strategies. Here, V20 denotes the Pascal VOC 20 dataset, and V20-C represents the average performance over its 15 synthetic corruption types. The variance in (a) is computed across all samples and all corruptions.

Adaptation (TTA) of OVSS models, which would enable models to dynamically adjust both to the task shift of VLM-based segmentation and to the domain shifts encountered during inference.

To close this gap, we present a novel **M**ulti-**L**evel **M**ulti-**P**rompt (MLMP) test time adaptation strategy, the first fully test-time adaptation framework that could be plugged into *any OVSS model*, to the best of our knowledge. MLMP is lightweight and plug-and-play, boosting performance on the fly without access to labels. Its power comes from two key ideas: (*i*) adaptively integrating intermediate vision-encoder layers to harvest complementary, shift-resilient features, and (*ii*) a multi-prompt optimization that exploits VLMs' template sensitivity to provide a robust adaptation signal across diverse text-template conditions.

The core requirement for test-time adaptation is a reliable signal that faithfully reflects the current input distribution—even under severe domain shifts or corruptions. To meet this need, MLMP begins by adaptively integrating intermediate layers of the vision encoder: earlier layers preserve fine-grained edges and textures, while deeper blocks encode semantic context, and each layer reacts differently when the data distribution changes. By aggregating these multi-level features into the adaptation process and weighting them by their confidence, MLMP harvests the most trustworthy signals for each input sample.

Beyond multi-level fusion, MLMP leverages VLMs' prompt sensitivity to model uncertainty. Prior work [5, 6] shows that changing a prompt template, e.g., from ''a photo of a {class}" to ''an origami of a {class}", could drastically change the classification performance. *In segmentation, we show that this effect is even more extreme: per-pixel predictions under different prompts diverge far more than the single CLS token used for classification (Figure 1a: CLS token vs. last-layer spatial tokens). This sensitivity effect is even more pronounced in intermediate feature maps. The intermediate layers that we fuse for reliability exhibit even stronger template-specific shifts (Figure 1a: intermediate-layer tokens).* Instead of viewing this inconsistency as a weakness, MLMP models it directly by incorporating multi-prompt, multi-level predictions into its adaptation objective function. This multi-prompt approach not only smooths out template-specific noise, thereby reducing gradient variance and preventing degenerate collapse, but also ensures that the model yields segmentations that are consistent across diverse linguistic formulations. In this way, MLMP transforms prompt sensitivity into a powerful adaptation signal that complements its multi-level feature integration. As illustrated in Figure 1b, our MLMP method consistently outperforms the non-adapted baseline, achieving about 8–9 absolute mIoU improvements on domain-shifted inputs, while also boosting performance on original (non-corrupted) images. This demonstrates the benefit of our joint multi-level, multi-prompt adaptation.

We outline our key contributions as follows:

- **Plug-and-Play TTA Framework for OVSS:** We introduce MLMP, which is, to the best of our knowledge, the first fully test-time adaptation method that could be easily applied to any OVSS backbone.
- **Adaptive Multi-Level Fusion:** MLMP integrates features from intermediate vision-encoder layers to capture complementary, shift-resilient cues. To further enhance robustness, we propose an uncertainty-aware strategy that re-weights features from individual layers based on their prediction entropy.
- **Multi-Prompt Local-Global Test-Time Optimization:** MLMP turns prompt sensitivity into signal by directly minimizing entropy across different text prompt templates at both the global CLS token and local pixel-wise levels. This optimization naturally complements our multi-level feature fusion by enforcing consistency across linguistic perspectives and feature depths.
- **Comprehensive OVSS TTA Benchmark Suite:** We curate a rigorous evaluation protocol spanning nine mainstream segmentation datasets and 15 common synthetic corruptions, and additional real and rendered domain shifts, **with a total of 87 distinct test scenarios**, establishing a standardized and comprehensive testbed for future TTA research in open-vocabulary segmentation. Our experiments on this suite demonstrate that MLMP consistently delivers significant gains over baselines across all scenarios.

## 2   Related Work

Test-time adaptation (TTA) for open-vocabulary semantic segmentation (OVSS) remains unexplored—existing TTA methods focus on classification or single-modality segmentation, while OVSS approaches use VLMs without any online adaptation. We bridge this gap with our proposed method MLMP, a plug-and-play TTA framework that can be applied to any OVSS method.

**Test-Time Adaptation.**  TTA addresses domain shifts by adapting pre-trained models to unlabeled target data without source samples. Methods like PTBN [7] and TENT [8] update batch statistics and affine parameters via entropy minimization but rely on large batches or augmentations. MEMO [9] simplifies this with single-sample augmentations, LAME [10] clusters features via Laplacian smoothing, and SAR [11] stabilizes adaptation using batch-agnostic normalization and sharpness-aware entropy minimization.

**Test-Time Adaptation on Segmentation.**  TTA enhances segmentation robustness against domain shifts without source data. Methods include self-supervised adaptation via entropy minimization or contrastive learning [8, 12], single-image adaptation optimizing per-image predictions [13], and continual TTA that leverages clustering to prevent forgetting [14]. Multi-modal adaptation uses cross-modal self-supervision [15], while active TTA integrates minimal human feedback for guided refinement [16]. These approaches assume a fixed, vision-only label space and rely on spatial or surrogate tasks, making them ill-suited for zero-shot, text-driven OVSS. Consequently, none have been applied to VLMs.

**Open-Vocabulary Semantic Segmentation.**  OVSS enables segmentation of unseen categories using vision-language models like CLIP. Approaches fall into fully-supervised, weakly-supervised, and training-free categories. Fully-supervised methods use pixel-wise annotations [17, 18, 19], while weakly-supervised ones leverage image-text pairs [20, 21, 22, 23]. Training-free OVSS avoids adaptation data but may rely on auxiliary pre-trained models [24, 25, 26]. Training-free OVSS approaches aim to enhance segmentation without additional training data. Some methods, such as SCLIP [27], adjust self-attention mechanisms to improve feature localization, while others, like MaskCLIP [28], refine feature extraction from CLIP's visual backbone. GEM [29] introduces additional optimization techniques to extract better dense features without fine-tuning. Among these, NACLIP [30] enhances CLIP's dense prediction capabilities by introducing neighborhood attention, which ensures that image patches focus on nearby regions, and by refining similarity measures to improve spatial consistency.

To the best of our knowledge, this is the first work to address TTA for OVSS models, filling a previously unexplored intersection between these fields.

To better situate this contribution within the broader landscape of general adaptation methods, Table 1 summarizes the key distinctions among zero-shot inference, domain generalization (DG) [31, 32], few-shot segmentation (FSS) [33], test-time training (TTT) [34, 35, 36], and our fully unsupervised test-time adaptation (TTA). This comparison highlights that MLMP addresses the most challenging

Table 1: Comparison of general learning and adaptation paradigms. Here, $x^s, y^s$ denote labeled source samples and $x^t, y^t$ denote target (test) samples and labels. Domain Generalization trains on labeled multi-domain source data to improve robustness on unseen targets, while few-shot and test-time training methods rely on labeled or source data during or after training. Our approach (Fully TTA) adapts solely using unlabeled test samples, requiring neither supervision nor source access.

| Setting | Source Data | Target Data | Train Loss | Test Loss |
|---|---|---|---|---|
| Zero-Shot Inference | ✗ | $x^t$ | ✗ | ✗ |
| Domain Generalization (DG) | $x^s, y^s$ (often multi-domain) | $x^t$ | $\mathcal{L}(x^s, y^s)$ (DG objectives) | ✗ |
| Few-Shot Learning (FSS) | ✗ | $x^t, y^t$ (few) | $\mathcal{L}(x^t, y^t)$ (FSS objectives) | ✗ |
| Test-Time Training (TTT) | $x^s, y^s$ | $x^t$ | $\mathcal{L}(x^s, y^s) + \mathcal{L}_{aux}(x^s)$ | $\mathcal{L}_{aux}(x^t)$ |
| **Fully Test-Time Adaptation (TTA, Ours)** | ✗ | $x^t$ | ✗ | $\mathcal{L}_{unsup}(x^t)$ |

and realistic scenario, adapting models entirely from unlabeled test data without any source access or annotated support samples.

# 3 Methodology

We first revisit the contrastive vision–language model (VLM) for open-vocabulary semantic segmentation (OVSS), and then present our Multi-Level Multi-Prompt (MLMP) adaptation strategy.

## 3.1 OVSS with VLMs

Given an input image $\mathbf{X} \in \mathbb{R}^{H \times W \times 3}$ and a set of concepts $C_k \in \mathcal{C}$ expressed in natural language, OVSS seeks a semantic mask $\mathbf{y} \in \{1, \ldots, K\}^{H \times W}$ that assigns one concept to every pixel.

Following recent approaches for OVSS [30, 27], we employ a transformer-based VLM to extract visual and text features from the image and concepts in natural language. Specifically, we feed the image $\mathbf{X}$ into the ViT-based vision encoder to extract a visual token matrix $\mathbf{F} = \left[ \mathbf{f}_{[\texttt{cls}]}, \mathbf{f}_1, \ldots, \mathbf{f}_N \right]$ with each $\mathbf{f}_i \in \mathbb{R}^D$, where $N = \lfloor H/s \rfloor \times \lfloor W/s \rfloor$ is the number of patches of size $s \times s$ in the image and $[\texttt{cls}]$ is the CLS token for classification. We define $\mathbf{Q} = \left[ \mathbf{q}_{[\texttt{cls}]}, \mathbf{q}_1, \ldots, \mathbf{q}_N \right]$, with each $\mathbf{q}_i \in \mathbb{R}^{D'}$, the output features before the projection layer: $\mathbf{F} = \text{proj}(\mathbf{Q})$. At the same time, the text encoder is employed to extract text features $\mathbf{t}_k \in \mathbb{R}^D$ for each concept $C_k \in \mathcal{C}$. This is achieved by combining $C_k$ with a text prompt template, for instance "A photo of a $[C_k]$" or "An image of a $[C_k]$" where $C_k$ is an arbitrary text description like "white horse".

The standard approach for classifying images with a contrastive VLM such as CLIP [1] computes the cosine between the CLS token features and text embeddings of classes, and assigns the image to the class with highest similarity:

$$\arg\max_k \ sim\left(\mathbf{f}_{[\texttt{cls}]}, \mathbf{t}_k\right), \quad \text{where } sim(\mathbf{x}, \mathbf{y}) = \frac{\mathbf{x} \cdot \mathbf{y}}{\|\mathbf{x}\|\|\mathbf{y}\|}. \tag{1}$$

For extending this approach to segmentation, we instead compute the similarity between *patch* embeddings $\mathbf{f}_i$ and text embeddings $\mathbf{t}_k$ and assign a class/concept to each patch.

## 3.2 MLMP: Proposed Method

Figure 2 illustrates our full test-time adaptation pipeline, **MLMP**. MLMP integrates three complementary ideas: **uncertainty-aware multi-level fusion**, **image-level entropy minimization** and **multi-prompt adaptation**.

We begin by modifying the entropy minimization objective of TENT [8] from image classification to work with *spatial tokens*. More specifically, for a batch of $B$ images, each containing $N$ tokens, the probability that token $i$ belongs to concept $k$ is

$$p_{ik} = \frac{\exp\left(sim(\mathbf{f}_i, \mathbf{t}_k)/\tau\right)}{\sum_{k'=1}^{|\mathcal{C}|} \exp\left(sim(\mathbf{f}_i, \mathbf{t}_{k'})/\tau\right)}. \tag{2}$$

where $\tau$ is a softmax temperature scaling parameter, $\mathbf{T} = \left[ \mathbf{t}_1, \ldots, \mathbf{t}_{|\mathcal{C}|} \right]$, and $\text{norm}(\cdot)$ denotes a function normalizing the columns of its input matrix to unit length. Also, let $\mathbf{P}$ denote a matrix

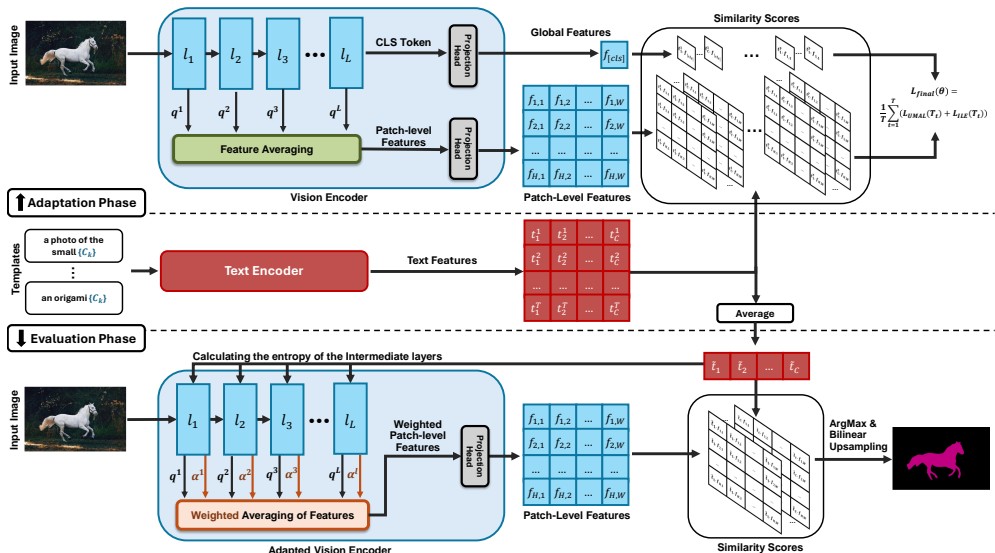

Figure 2: Overview of our MLMP method. In the Adaptation Phase, the model is adapted by leveraging multiple prompt templates alongside various intermediate feature layers, as well as the global feature. During the Evaluation Phase, the model computes weights based on the entropy of the intermediate features to perform a weighted averaging. These averaged features, combined with the different templates, are then used to generate the final segmentation map.

containing the probabilities in (2), which could be expressed more compactly as follows:

$$\mathbf{P} = \text{softmax}\big(\text{norm}(\mathbf{F}) \cdot \text{norm}(\mathbf{T})^{\top}/\tau\big). \tag{3}$$

The batch-wise entropy, which is minimized for adaptation, is then defined as follows:

$$\mathcal{H}(\mathbf{P}) = -\frac{1}{B \cdot N} \sum_{i=1}^{B \cdot N} \sum_{k=1}^{|\mathcal{C}|} p_{ik} \log p_{ik}. \tag{4}$$

Following [5, 37], we keep the entire text encoder frozen and update only the *LayerNorm* parameters of the vision encoder during adaptation. Freezing the text encoder greatly reduces computational overhead, since text embeddings can be precomputed and reused across all test samples.

**Uncertainty-Aware Multi-Level Fusion.** In VLM-based classification, the CLS token in the last layer of the visual encoder is typically used to compute the class label probabilities. This approach relies on the idea that the relevant information for classification lies at the end of the ViT and that intermediate layers serve to transform features. In segmentation, however, features from intermediate layers are often used to capture complementary information at different scales [38]. This hypothesis is validated in Table 2 as well as Figure 3, showing that a higher segmentation mIoU is obtained when combining the features from different intermediate layers.

Inspired by this result, and leveraging the useful property of ViTs that the output of each layer has the same shape, we extend the entropy-based loss described above to use features from *multiple layers*. Denoting as $\mathbf{q}_i^{\ell}$ the visual features of patch $i$ obtained at layer $\ell$, we seek to aggregate the multi-level features into a single vector $\overline{\mathbf{q}}_i$ for segmentation prediction. A simple approach for doing this is to compute $\overline{\mathbf{q}}_i$ by averaging $\mathbf{q}_i^{\ell}$ across all layers $\ell$. However, this approach ignores the relative contribution and confidence of each layer in the final segmentation. To address this limitation, we estimate a confidence weight $\alpha^{\ell}$ for each layer $\ell$ based on its prediction entropy. First, we get visual features $\mathbf{F}_i^{\ell} = \text{proj}(\mathbf{Q}_i^{\ell})$ using the *same* projection head as for the final segmentation. Following the same approach as before, we then compute the batch-wise entropy of layer $\ell$ as

$$h^{\ell} = \mathcal{H}(\mathbf{P}^{\ell}), \quad \text{with } \mathbf{P}^{\ell} = \text{softmax}\big(\text{norm}(\mathbf{F}^{\ell}) \cdot \text{norm}(\mathbf{T})^{\top}/\tau\big). \tag{5}$$

Finally, the confidence weight of the layer is obtained using a softmax as follows:

$$\alpha^{\ell} = \frac{\exp(-\beta \cdot h^{\ell})}{\sum_{\ell'=1}^{L} \exp(-\beta \cdot h^{\ell'})}. \tag{6}$$

Here, $\beta$ is a parameter controlling the "sharpness" of the weight distribution. During adaptation, we set $\beta = 0$ to promote a uniform contribution from all layers in the prediction. During inference, we sharpen the distribution with a value of $\beta = 1$, emphasizing the more confident layers in the final prediction.

With these confidence weights, we can now obtain our uncertainty-aware multi-level (UAML) features as

$$\overline{\mathbf{F}} = \text{proj}(\overline{\mathbf{Q}}), \text{ with } \overline{\mathbf{Q}} = \sum_{\ell=1}^{L} \alpha^\ell \mathbf{Q}^\ell, \tag{7}$$

giving the following entropy-based loss to minimize:

$$\mathcal{L}_{\text{UAML}}(\mathbf{T}) = \mathcal{H}(\overline{\mathbf{P}}), \text{ with } \overline{\mathbf{P}} = \text{softmax}\big(\text{norm}(\overline{\mathbf{F}}) \cdot \text{norm}(\mathbf{T})^\top / \tau\big). \tag{8}$$

**Image-Level Entropy Minimization.** Since the CLS token is not directly linked to individual patch predictions but rather captures a more global representation of the input, we also include an image-level entropy (ILE) minimization term specifically for this token. As illustrated in Figure 1, the CLS token demonstrates increased robustness and reliability. This term, which encourages the model to produce more confident global predictions is expressed as:

$$\mathcal{L}_{\text{ILE}}(\mathbf{T}) = -\frac{1}{B} \sum_{b=1}^{B} \sum_{k=1}^{|\mathcal{C}|} p_{b,k}^{[\text{cls}]} \log p_{b,k}^{[\text{cls}]}. \tag{9}$$

Here, $p_{b,k}^{[\text{cls}]}$ denotes the predicted probability for concept $C_k$ obtained using the CLS token in the last layer for the $b$-th sample.

**Multi-Prompt Adaptation.** Prior work on TTA for classification [5] has shown the usefulness of leveraging multiple prompt templates in VLM to encode class labels, based on the idea that the templates capture complementary information about these classes. As shown in Figure 1a, the sensitivity to the choice of prompt templates is even more pronounced in segmentation tasks, where fine-grained spatial predictions are required. Using multiple templates acts as cross-modal regularization, encouraging more stable and generalized learning signals. While different from image augmentation, it can be seen as a strong, safe, and lightweight text-space augmentation. Rather than averaging the weights adapted from different prompt templates as in [5]—a computationally expensive approach for dense prediction tasks such as segmentation—our method minimizes our proposed UAML and ILE losses across these templates. Let $\mathbf{T}_t$ be the text features obtained using the $t$-th template. Our final adaptation loss is defined as:

$$\mathcal{L}_{\text{final}}(\theta) = \frac{1}{T} \sum_{t=1}^{T} (\mathcal{L}_{\text{UAML}}(\mathbf{T}_t) + \mathcal{L}_{\text{ILE}}(\mathbf{T}_t)). \tag{10}$$

**Theoretical Justification.** Each template $t$ contributes its own adaptation loss, as we optimize their average in Eq. (10). By optimizing the adaptation loss of each prompt directly, we force the model to correct for the unique wording and visual cue of each template, rather than 'averaging' these differences in the text embedding space. This loss-level integration treats each template as an independent critic, translating diverse linguistic perspectives into separate gradient signals. Averaging those signals produces an unbiased descent direction whose variance decays as $1/T$, enabling each adaptation step to represent the full prompt ensemble while being stable under noisy shifts.

**Proposition 1 (Unbiasedness and Variance Bound).** Assume that each per-template gradient $g_t(\theta) = \nabla_\theta\big[\mathcal{L}_{\text{UAML}}(\mathbf{T}_t) + \mathcal{L}_{\text{ILE}}(\mathbf{T}_t)\big]$ has variance bounded by $\sigma^2$, then the ensemble gradient, defined by $\nabla_\theta \mathcal{L}_{\text{final}} = \frac{1}{T} \sum_{t=1}^{T} g_t(\theta)$, is unbiased and satisfies the following variance bound:

$$\mathbb{E}\Big[\nabla_\theta \mathcal{L}_{\text{final}}\Big] = \mathbb{E}\big[g_t(\theta)\big]; \quad \text{Var}\Big(\nabla_\theta \mathcal{L}_{\text{final}}\Big) = \frac{1}{T^2} \sum_{t=1}^{T} \text{Var}\big(g_t(\theta)\big) \le \frac{\sigma^2}{T} \tag{11}$$

*Proof.* A proof of Prop. 1 is provided in the Appendix. $\square$

The $1/T$ reduction in gradient variance, as stated in Prop. 1, explains the improved stability we observe. Table 4 confirms that this loss-level ensemble outperforms alternative fusion strategies.

Table 2: mIoU performance when using different layer ranges in the proposed multi-level adaptation.

| ViT-L/14 Layer Range | $L_{24}$ (last) | $L_{23-24}$ (last two) | $L_{22-24}$ (last three) | $L_{19-24}$ (last 25%) | $L_{13-24}$ (last 50%) | $L_{7-24}$ (last 75%) | $L_{1-24}$ (all layers) |
|---|---|---|---|---|---|---|---|
| V20 (Original) | $77.00_{\pm0.04}$ | $77.65_{\pm0.02}$ | $77.66_{\pm0.09}$ | $80.61_{\pm0.05}$ | $80.50_{\pm0.03}$ | $\mathbf{81.67}_{\pm0.04}$ | $78.79_{\pm0.02}$ |
| Gaussian Noise | $63.02_{\pm0.06}$ | $64.41_{\pm0.05}$ | $65.39_{\pm0.13}$ | $66.88_{\pm0.18}$ | $66.88_{\pm0.02}$ | $\mathbf{67.82}_{\pm0.01}$ | $63.06_{\pm0.09}$ |
| Defocus Blur | $72.06_{\pm0.12}$ | $72.93_{\pm0.19}$ | $72.84_{\pm0.02}$ | $76.10_{\pm0.16}$ | $76.37_{\pm0.05}$ | $\mathbf{78.78}_{\pm0.02}$ | $77.56_{\pm0.09}$ |
| Snow | $71.04_{\pm0.05}$ | $72.09_{\pm0.04}$ | $72.56_{\pm0.02}$ | $74.47_{\pm0.12}$ | $74.41_{\pm0.01}$ | $\mathbf{76.39}_{\pm0.02}$ | $73.72_{\pm0.07}$ |
| JPEG Compression | $71.84_{\pm0.15}$ | $73.88_{\pm0.11}$ | $74.40_{\pm0.07}$ | $76.96_{\pm0.02}$ | $77.67_{\pm0.03}$ | $\mathbf{78.73}_{\pm0.08}$ | $75.87_{\pm0.19}$ |
| V20-C Average | 69.33 | 70.33 | 70.78 | 72.89 | 73.45 | **74.90** | 72.02 |

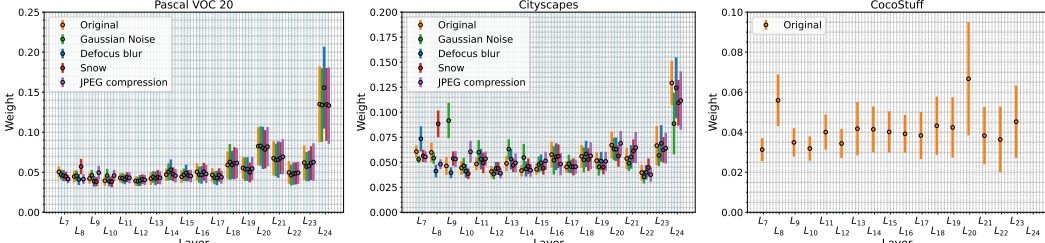

Figure 3: Mean and standard deviation of layer-wise confidence weights of MLMP across datasets. The fusion mechanism adaptively emphasizes more reliable layers based on input conditions.

# 4 Experimental Settings

**Experimental Setup.** Following prior work on TTA in classification [37, 5], we restrict updates to the normalization layers within the vision encoder. The adaptation process is carried out over 10 iterations using the Adam optimizer with a constant learning rate of $10^{-3}$ across all datasets. We use a batch size of 2 images during adaptation across all datasets. For each new batch, the model undergoes a reset, restoring it to its initial weights before adaptation is applied.

**Datasets.** In traditional TTA for segmentation, two datasets are commonly employed to simulate domain shifts—one for model training and another for adaptation during inference (e.g., GTAV and Cityscapes)—as both must share the same semantic label space. In our study, as this is the first exploration of TTA for VLMs in segmentation tasks and given that VLMs are pre-trained, we draw inspiration from ImageNet-C [39] to introduce 15 synthetic corruptions on segmentation datasets. Our experiments are conducted on Pascal VOC 20 (v20), Pascal VOC 21 (v21) [40], Pascal Context 59 (P59), Pascal Context 60 (P60) [41], and Cityscapes [42], incorporating both original version (clean) and the synthetic 15 corruptions (denoted with a "-C" suffix). For COCO-Stuff [43] and COCO-Object [44], we use only the original versions. To further evaluate robustness under real and rendered distributional shifts, we additionally include ACDC [45]—capturing real-world adverse conditions such as fog, night, rain, and snow—and GTA-V [46], which provides photorealistic, game-rendered urban scenes. This extended setup results in **87 distinct test scenarios** encompassing synthetic, real, and rendered shifts, enabling a comprehensive evaluation of MLMP across diverse conditions.

**Benchmarking.** While MLMP is compatible with any OVSS framework, we incorporate NA-CLIP [30] with ViT-L/14 as our baseline OVSS model, which leverages neighborhood attention to enhance spatial consistency in a training-free manner. The compared methods include TENT [8], which serves as a baseline and minimizes entropy during adaptation; CLIPArTT [37], which employs pseudo-labels generated via conformal learning; WATT [5], which averages learnable parameters across multiple parallel branches; and TPT [6], which performs prompt tuning to adapt VLMs at test time. For a fair comparison, we modified all methods for the segmentation setting by processing all spatial tokens extracted from the VLM, rather than relying solely on the CLS token.

# 5 Results

## 5.1 Ablation studies

**Effect of Intermediate Layers.** To analyze the impact of layer selection in our uncertainty-aware multi-level adaptation strategy, we use different ranges of intermediate layers. As shown in Table 2,

Table 3: mIoU comparison of MLMP components, showing individual and combined contributions.

| Multi-Level Fusion | ✗ | ✓ | ✓ | ✗ | ✗ | ✓ | ✓ | ✓ | ✓ | ✗ | ✓ | ✓ |
|---|---|---|---|---|---|---|---|---|---|---|---|---|
| Multi-Prompt Loss | ✗ | ✗ | ✗ | ✓ | ✗ | ✓ | ✓ | ✗ | ✗ | ✓ | ✓ | ✓ |
| Image-Level Entropy | ✗ | ✗ | ✗ | ✗ | ✓ | ✗ | ✗ | ✓ | ✓ | ✓ | ✓ | ✓ |
| Uncertainty-Aware Weighting | ✗ | ✗ | ✓ | ✗ | ✗ | ✗ | ✓ | ✗ | ✓ | ✗ | ✗ | ✓ |
| V20 (Original) | 77.00 | 77.38 | 81.67 | 79.70 | 78.74 | 78.97 | 83.00 | 77.69 | 82.70 | 81.15 | 79.13 | **83.76** |
| Gaussian Noise | 63.02 | 65.42 | 67.82 | 66.75 | 65.66 | 65.96 | 69.13 | 66.17 | 69.00 | 69.62 | 67.35 | **71.13** |
| Defocus Blur | 72.06 | 76.65 | 78.78 | 74.31 | 75.00 | 76.46 | 78.78 | 77.29 | 79.78 | 77.14 | 77.79 | **80.36** |
| Snow | 71.04 | 72.64 | 76.39 | 74.66 | 74.16 | 73.25 | 77.31 | 74.05 | 78.50 | 77.20 | 74.94 | **79.53** |
| JPEG Compression | 71.84 | 74.38 | 78.73 | 75.56 | 74.77 | 76.77 | 80.81 | 74.61 | 79.79 | 77.98 | 77.94 | **82.06** |
| V20-C Average | 69.33 | 71.59 | 74.90 | 72.58 | 71.99 | 72.66 | 75.97 | 72.41 | 76.18 | 75.08 | 73.89 | **77.58** |

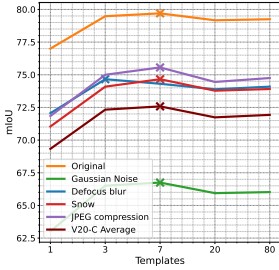

Figure 4: mIoU performance of our method for different numbers of templates.

Table 4: mIoU performance for prompt-integration strategies (Text, Params, Loss) on clean and corrupted data.

| Dataset: V20 | Text | Params | Loss |
|---|---|---|---|
| Original | 78.91 ±0.07 | 74.46 ±0.21 | **79.70 ±0.06** |
| Gaussian Noise | 66.27 ±0.00 | 62.83 ±0.04 | **66.75 ±0.01** |
| Defocus Blur | 74.05 ±0.10 | 70.28 ±0.16 | **74.31 ±0.09** |
| Snow | 73.78 ±0.02 | 70.10 ±0.30 | **74.66 ±0.01** |
| JPEG Compression | 74.98 ±0.05 | 70.55 ±0.11 | **75.56 ±0.02** |
| V20-C Average | 71.92 | 68.44 | **72.58** |

performance varies notably with the fusion range. While using only the final layers yields moderate improvements, incorporating the last 75% of the layers consistently achieves the best performance across both clean and corrupted inputs. This highlights that multi-level fusion is a key driver of adaptation performance: earlier layers, although less semantically abstract, contribute valuable low-level features—such as texture and edge cues—that enhance robustness to distribution shifts. In this ablation, we isolate the effect of multi-level fusion by applying only the first term in Eq. 10, using a single prompt template and omitting the $L_{ILE}$ term.

**Effect of Uncertainty-Aware Layer Fusion.** We investigate strategies for aggregating multi-level features by comparing uniform averaging ($\beta = 0$) and uncertainty-aware fusion ($\beta = 1$) during evaluation, using the same 75% layer range identified in the previous ablation. As shown in Table 3, incorporating entropy-based weighting improves performance by 4.29% on V20 and 3.31% on V20-C. This highlights the importance of leveraging layer-wise confidence when aggregating features.

**Visualization of Layer-Wise Confidence Weights.** We visualize the mean and standard deviation of the learned layer weights to better understand the behavior of our uncertainty-aware fusion strategy. As shown in Figure 3, deeper layers tend to receive higher confidence, though earlier layers also contribute, especially under corrupted conditions. This variation is most pronounced in the Cityscapes dataset, where the distribution fluctuates more across layers and corruption types. In contrast, the COCO-Stuff dataset shows a flatter distribution, where the final layer is not consistently the most influential. These results underscore the core strength of our fusion mechanism: its ability to adaptively reweight layers based on input conditions, assigning greater importance to those that remain more reliable under distribution shifts and corruption. Please refer to the Appendix for additional results on other datasets.

**Effect of Global Image-Level Adaptation Term.** The image-level entropy term, $L_{ILE}$ complements our patch-level adaptation by encouraging consistent global predictions through the CLS token. While the multi-level loss targets fine-grained spatial predictions, the ILE term introduces global context that helps stabilize adaptation. As shown in Table 3, when added in isolation, $L_{ILE}$ improves performance by 1.74% and 2.66% on V20 and V20-C, respectively, demonstrating the benefit of incorporating global context under distribution shift.

**Effect of Number of Prompt Templates.** To isolate this effect, we evaluate performance using different numbers of prompt templates while disabling both the multi-level fusion and the image-level entropy (ILE) term in Eq.10. As shown in Figure4, increasing the number of templates improves performance up to 7, after which the gains begin to saturate or slightly decline. This trend holds across

Table 5: mIoU comparison of MLMP and baselines across several datasets. CLIPArTT could not be run for a few cases owing to GPU memory shortages. Full per-dataset results are in the Appendix.

| OVSS Backbone: NACLIP | | Adaptation Method | | | | |
|---|---|---|---|---|---|---|
| Dataset | No Adapt. | TENT | TPT | WATT | CLIPArTT | MLMP |
| V20 (Original) | 75.91 | 77.00 ±0.04 | 75.93 ±0.01 | 57.73 ±0.06 | 72.77 ±0.14 | **83.76 ±0.00** |
| V20-C  Gaussian Noise | 62.89 | 63.02 ±0.06 | 62.98 ±0.01 | 36.44 ±0.04 | 53.36 ±0.25 | **71.13 ±0.09** |
| Shot noise | 66.26 | 65.88 ±0.06 | 66.33 ±0.02 | 40.95 ±0.05 | 58.15 ±0.28 | **75.02 ±0.03** |
| Impulse Noise | 63.16 | 64.17 ±0.04 | 63.12 ±0.01 | 34.90 ±0.06 | 54.83 ±0.03 | **71.34 ±0.11** |
| Defocus blur | 72.59 | 72.06 ±0.12 | 72.55 ±0.02 | 52.43 ±0.03 | 65.39 ±0.45 | **80.36 ±0.06** |
| Glass blur | 71.44 | 70.74 ±0.07 | 71.40 ±0.01 | 49.96 ±0.05 | 64.62 ±0.13 | **78.84 ±0.05** |
| Motion blur | 73.10 | 73.50 ±0.10 | 73.16 ±0.02 | 53.35 ±0.06 | 67.48 ±0.17 | **81.41 ±0.05** |
| Zoom blur | 59.03 | 61.36 ±0.07 | 59.00 ±0.01 | 41.39 ±0.08 | 52.37 ±0.12 | **69.41 ±0.12** |
| Snow | 71.49 | 71.04 ±0.05 | 71.44 ±0.01 | 51.18 ±0.06 | 66.97 ±0.02 | **79.53 ±0.05** |
| Frost | 65.38 | 67.01 ±0.02 | 65.46 ±0.01 | 45.75 ±0.05 | 60.48 ±0.08 | **73.20 ±0.07** |
| Fog | 70.69 | 70.54 ±0.07 | 70.70 ±0.01 | 52.96 ±0.04 | 67.85 ±0.10 | **79.81 ±0.06** |
| Brightness | 74.95 | 75.61 ±0.02 | 74.95 ±0.01 | 55.82 ±0.05 | 71.52 ±0.14 | **83.51 ±0.01** |
| Contrast | 71.51 | 70.51 ±0.04 | 71.49 ±0.02 | 50.74 ±0.06 | 66.01 ±0.06 | **79.06 ±0.16** |
| Elastic transform | 62.86 | 65.78 ±0.05 | 62.95 ±0.01 | 45.45 ±0.04 | 60.41 ±0.10 | **74.03 ±0.01** |
| Pixelate | 77.28 | 76.95 ±0.12 | 77.31 ±0.01 | 59.76 ±0.05 | 73.14 ±0.17 | **84.97 ±0.04** |
| JPEG compression | 72.59 | 71.84 ±0.15 | 72.56 ±0.01 | 53.44 ±0.05 | 68.21 ±0.07 | **82.06 ±0.01** |
| Average | 69.01 | 69.33 | 69.03 | 48.30 | 63.39 | **77.58** |
| V21 (Original) | 45.12 | 45.65 ±0.02 | 45.17 ±0.01 | 28.58 ±0.05 | 39.50 ±0.04 | **50.78 ±0.02** |
| V21-C Average | 40.75 | 40.95 | 40.77 | 24.12 | 34.16 | **46.25** |
| P59 (Original) | 28.23 | 28.73 ±0.02 | 28.26 ±0.01 | 16.55 ±0.04 | 24.60 ±0.03 | **31.95 ±0.02** |
| P59-C Average | 23.88 | 23.88 | 23.88 | 13.37 | 19.72 | **27.03** |
| P60 (Original) | 24.95 | 25.29 ±0.01 | 24.98 ±0.01 | 14.77 ±0.03 | 21.88 ±0.03 | **27.99 ±0.03** |
| P60-C Average | 21.39 | 21.25 | 21.49 | 12.08 | 17.79 | **24.07** |
| CityScapes (Original) | 29.49 | 30.54 ±0.04 | 29.57 ±0.01 | 20.77 ±0.06 | – | **33.35 ±0.03** |
| CityScapes-C Average | 21.63 | 21.64 | 21.60 | 13.45 | – | **23.02** |
| COCOObject (Original) | 23.80 | 24.88 ±0.01 | 23.84 ±0.01 | 14.14 ±0.06 | 21.34 ±0.03 | **28.84 ±0.01** |
| COCOStuff (Original) | 18.34 | 18.76 ±0.01 | 18.35 ±0.01 | 9.49 ±0.02 | 15.48 ±0.01 | **21.25 ±0.01** |

both clean and corrupted settings, indicating that a moderate number of diverse prompt templates is sufficient for MLMP. We therefore use 7 templates by default in our main experiments. Please refer to the Appendix for template details.

**Where to Integrate Multi-Prompt Information.** Here, we empirically compare strategies for integrating multi-prompt information into the adaptation process. Specifically, we evaluate: (1) text-level averaging, where prompt embeddings are averaged before computing logits—a technique commonly used in zero-shot learning [1]; (2) a learnable parameter averaging baseline (Params) inspired by WATT [5]; and (3) our proposed method (Loss), which incorporates all prompt templates directly into the adaptation loss (Eq.10). To isolate the effect of prompt integration, this analysis excludes other components such as multi-level fusion and the image-level entropy (ILE) term. As shown in Table 4, our loss-level formulation consistently outperforms the alternatives across both clean and corrupted settings.

**Full Component Analysis.** Table 3 presents an extensive ablation evaluating the contribution of each component in our MLMP strategy. Each proposed element yields consistent gains when added independently, but it is their combination that delivers the highest overall performance across both clean and corrupted settings, highlighting their strong complementary effects within a unified framework. Additionally, we provide further ablations in the Appendix, including alternative OVSS backbones, different VLMs, ViT architecture variants, computational complexity analysis, and MLMP segmentation map visualizations, as well as discussions on the effect of longer prompts, effect of adaptation iterations, and episodic vs. online adaptation.

## 5.2 Final Comparison with Alternative Adaptation Methods

**Performance on Clean Data (No Distributional Shift).** We begin by evaluating MLMP on clean test data (original), where no distributional shift is present. This setting is crucial, as TTA methods must avoid degrading performance when adaptation is unnecessary. As shown in Table 5, MLMP achieves strong mIoU gains of +7.85, +5.66, +3.72, and +3.04 on V20, V21, P59, and P60, and

+5.04/+2.91 on challenging datasets COCOObject and COCOStuff. In contrast, most alternative adaptation methods fail to improve performance in this setting. These gains highlight the robustness and generalization of our method, even when no explicit domain shift is present.

**Performance Under Distributional Shift.** Under distributional shifts, the advantages of MLMP become even more apparent. As shown in Table 5, MLMP consistently outperforms both the zero-shot baseline and existing adaptation methods, achieving mIoU gains of +8.60, +5.50, +3.15, and +2.68 on V20-C, V21-C, P59-C, and P60-C, respectively. Beyond standard corruptions, we further evaluate on the Cityscapes dataset, which presents natural domain shifts such as environmental variation, weather conditions, and resolution differences. Despite its challenging nature and low zero-shot performance, MLMP improves mIoU by +3.86, demonstrating its real-world adaptability. To push this further, we apply 15 corruption types to create Cityscapes-C, where MLMP still yields a +1.39 gain. While TENT provides modest improvements, most other adaptation methods—including ClipArTT, TPT, and WATT—either fail to improve or degrade performance. These results highlight that naive strategies like pseudo-labeling, prompt tuning, or weight averaging are insufficient for open-vocabulary segmentation, and emphasize the need for segmentation-specific adaptation techniques such as MLMP. Detailed results can be found in the Appendix.

While Cityscapes already embodies natural distributional shifts caused by variations in lighting, camera viewpoint, and urban layouts, we further assess MLMP on datasets explicitly designed to capture targeted domain shifts. Specifically, we evaluate on ACDC [45], which contains real-world adverse conditions (Fog, Night, Rain, and Snow), and GTA-V [46], a photorealistic, game-rendered dataset that introduces a distinct synthetic distribution shift relative to real imagery seen during CLIP pre-training. As summarized in Table 6, MLMP achieves consistent improvements over both the non-adapted baseline and TENT across all ACDC domains, yielding on average +6 mIoU gains under real-world conditions. Similarly, on the GTA-V dataset, MLMP improves by +3.8 mIoU over the baseline, further confirming its robustness across both realistic and rendered distribution shifts.

Table 6: mIoU comparison on realistic (ACDC) and rendered (GTA-V) domain shifts. Full ACDC results (including reference/clean views) are provided in the Appendix.

| OVSS: NACLIP | | Adaptation Method | | |
|---|---|---|---|---|
| Dataset | | No Adapt. | TENT | MLMP |
| ACDC | Fog | 23.88 | 26.89 ±0.04 | **33.33** ±0.04 |
| | Night | 22.12 | 24.17 ±0.00 | **24.76** ±0.03 |
| | Rain | 23.86 | 26.84 ±0.04 | **32.44** ±0.04 |
| | Snow | 23.54 | 27.25 ±0.05 | **30.59** ±0.03 |
| | Average | 23.35 | 26.29 | **30.28** |
| GTA-V | | 25.09 | 26.62 ±0.01 | **28.84** ±0.02 |

## 6 Conclusion

We presented MLMP, a plug-and-play test-time adaptation framework for open-vocabulary semantic segmentation that can be integrated with any OVSS method. By combining uncertainty-aware fusion of intermediate ViT features with a novel loss-level integration of multiple prompt templates, MLMP consistently enhances performance across both clean and shifted domains—including common corruptions and natural distributional shifts. Our comprehensive OVSS-TTA benchmark—covering nine datasets and 87 distinct test scenarios—demonstrates MLMP's broad applicability and establishes a rigorous evaluation protocol for future work in adaptive, language-aware segmentation. While MLMP demonstrates strong, consistent gains, there remain opportunities to further refine its components. In particular, our current layer-weighting mechanism relies on entropy estimates from a shared projection head, which may not fully reflect each layer's unique characteristics. Future work could investigate more flexible architectures—such as lightweight adapters or dedicated projection modules per layer—to more accurately assess and fuse intermediate features.

## Acknowledgments

We appreciate the computational resources and support provided by Compute Canada and the Digital Research Alliance of Canada.

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

# Test-Time Adaptation of Vision-Language Models for Open-Vocabulary Semantic Segmentation - Appendix

## A  Proof of Proposition 1

**Unbiasedness and Variance Bound Proposition 1 (Restated):** Assume each per-template gradient $g_t(\theta) = \nabla_\theta[L_{\text{UAML}}(T_t) + L_{\text{ILE}}(T_t)]$ has variance bounded by $\sigma^2$. Then, the ensemble gradient, defined by $\nabla_\theta \mathcal{L}_{\text{final}} = \frac{1}{T}\sum_{t=1}^{T} g_t(\theta)$, is unbiased and satisfies the following variance bound:

$$\mathbb{E}[\nabla_\theta L_{\text{final}}] = \mathbb{E}[g_t(\theta)], \quad \text{Var}(\nabla_\theta L_{\text{final}}) = \frac{1}{T^2}\sum_{t=1}^{T}\text{Var}(g_t(\theta)) \leq \frac{\sigma^2}{T}.$$

**Step 1: Unbiasedness.** By linearity of expectation, we have:

$$\mathbb{E}\left[\nabla_\theta L_{\text{final}}\right] = \mathbb{E}\left[\frac{1}{T}\sum_{t=1}^{T} g_t(\theta)\right] = \frac{1}{T}\sum_{t=1}^{T}\mathbb{E}[g_t(\theta)] = \mathbb{E}[g_t(\theta)].$$

Hence, averaging the gradients from all $T$ templates gives an unbiased estimate of the true gradient of the final loss.

**Step 2: Variance Bound.** Assuming independence with $\text{Var}(g_t) \leq \sigma^2$, we get

$$\text{Var}(\nabla_\theta L_{\text{final}}) = \text{Var}\left(\frac{1}{T}\sum_{t=1}^{T} g_t(\theta)\right) = \frac{1}{T^2}\sum_{t=1}^{T}\text{Var}(g_t(\theta)) \leq \frac{T\sigma^2}{T^2} = \frac{\sigma^2}{T}$$

Thus, the variance bound holds:

$$\text{Var}(\nabla_\theta L_{\text{final}}) \leq \frac{\sigma^2}{T}.$$

This completes the proof of Proposition 1.

## B  Implementation Details

Unless otherwise noted, all experiments utilize NACLIP [30] with ViT-L/14 as the OVSS backbone. We adapt only the LayerNorm parameters within the vision encoder, amounting to approximately 0.02% of the model's total parameters. Our adaptation setup follows prior work in classification [5, 37], using the Adam optimizer with a fixed learning rate of 0.001 and 10 adaptation steps (iterations) across all experiments. Additionally we use batch of 2 images during adaptation. After each batch, model weights are explicitly reset to their original pre-adaptation values to ensure that each batch is adapted independently, without leveraging information from previously processed data. Following standard settings from [1], we use the default softmax temperature value of 100 in all experiments. All images are resized to $224 \times 224$ pixels. Due to the high resolution of images in the Cityscapes dataset, we split them into overlapping patches of size $224 \times 224$ pixels with an overlap of 112 pixels between patches. The segmentation predictions from these patches are aggregated to reconstruct the final, full-resolution segmentation maps. No image augmentation techniques are applied during either the adaptation or evaluation phases.

All experiments are conducted on NVIDIA V100 GPUs equipped with 32GB memory. We implement our approach using the PyTorch deep learning framework. To ensure statistical robustness and fairness in our comparisons, we repeat each experiment three times, reporting average performance along with standard deviation. We have provided detailed instructions and step-by-step scripts in our repository, clearly demonstrating how to generate datasets, perform the described test-time adaptations, and reproduce our reported results.

We adapt other baseline methods (TENT [8], TPT [6], WATT [5], CLIPArTT [37]) to work with spatial tokens. General adaptation hyperparameters (optimizer, learning rate, adaptation steps, and

Table 7: List of the seven prompt templates used in our MLMP method. These general-purpose templates, originally proposed by the CLIP authors, serve as diverse textual views of each class and are not tailored to specific datasets or domains.

| ID | Prompt template |
|---|---|
| $T^1$ | `itap of a {class}` |
| $T^2$ | `a bad photo of the {class}.` |
| $T^3$ | `a origami {class}.` |
| $T^4$ | `a photo of the large {class}.` |
| $T^5$ | `a {class} in a video game.` |
| $T^6$ | `art of the {class}.` |
| $T^7$ | `a photo of the small {class}.` |

batch size) remain consistent with our setup. Method-specific hyperparameters or components are retained as reported in their original implementations. Specifically, for WATT, we used the sequential version (WATT-S) with default values of $l = 2$ and $m = 5$, for CLIPArTT we used the default $k = 3$, and for TPT we used the 4 learnable tokens. Adapting these baseline methods to the segmentation task allowed us to systematically evaluate how various adaptation strategies—such as prompt tuning (TPT), pseudo-labeling (WATT, CLIPArTT), prompt refinement (CLIPArTT), and weight averaging (WATT)—perform in the context of open-vocabulary segmentation.

## C  Template Details

Table 7 lists the seven prompt templates used in our MLMP method. These templates were selected by the original CLIP authors[2] and are general-purpose, not tailored to any specific image content. The CLIP repository also provides a full set of 80 prompt templates, which we use for larger-scale ablations.

More specifically, for the ablation in Figure 4 of the main paper, we vary the number of templates $T$ as follows:

- $T = 1$: the default CLIP prompt "`a photo of a {class}`"
- $T = 3$: the first three templates $\{T^1, T^2, T^3\}$ in Table 7
- $T = 7$: all seven templates $\{T^1, \ldots, T^7\}$ in Table 7
- $T = 20$: the first 20 templates from the 80-template pool
- $T = 80$: the complete set of 80 CLIP templates

## D  Dataset Details

This section details the datasets used in our experiments, along with the synthetic, real, and rendered domain shifts applied to evaluate robustness under distributional changes.

More specifically, we conduct experiments on a diverse set of segmentation benchmarks:

- **Pascal VOC (V20/V21)** [40]: Contains 20 foreground object classes (v20) with a background class (v21), widely used for benchmarking semantic segmentation tasks.
- **Pascal Context (P59/P60)** [41]: An extension of the Pascal VOC 2010 dataset, providing pixel-level annotations for more than 400 classes. Due to sparsity, a frequently used subset includes 59 object classes plus a background class, totaling 60 categories.
- **Cityscapes** [42]: A large-scale dataset for semantic segmentation of urban street scenes. It comprises around 5,000 finely annotated images from 50 cities, recorded under various daylight conditions, featuring dynamic objects, varying layouts, and changing backgrounds, capturing significant natural domain shifts.

---

[2]`https://github.com/openai/CLIP/blob/main/notebooks/Interacting_with_CLIP.ipynb`

- **COCO-Stuff** [43]: Extends the original COCO dataset by adding annotations for background categories ("stuff"), resulting in 171 classes—80 objects, and 91 stuff categories.

- **COCO-Object** [44]: A subset of the original COCO dataset, consisting exclusively of 80 object categories without background annotations.

- **ACDC** [45]: Includes pixel-level annotations for 19 semantic classes, covering diverse driving scenes under fog, night, rain and snow conditions.

- **GTA-V** [46]: Contains 24,966 synthetic images with pixel-accurate semantic labels; one version aligns to 34/19 classes compatible with real-world segmentation benchmarks.

In addition to the original versions of each dataset—some of which already reflect natural domain shifts (e.g., Cityscapes)—we further assess the robustness of all methods by applying synthetic corruptions. Inspired by ImageNet-C [39], we apply **15 synthetic corruptions** to evaluate the robustness of segmentation models under various perturbations. These corruptions include Gaussian noise, shot noise, impulse noise, defocus blur, glass blur, motion blur, zoom blur, snow, frost, fog, brightness variations, contrast variations, elastic transformations, pixelation, and JPEG compression. Each corruption is applied at severity level 5, representing the most challenging scenario.

We resize images from all datasets to $224 \times 224$ pixels. Due to the high resolution of Cityscapes and ACDC images, we process them as overlapping patches of size $224 \times 224$ pixels (with overlaps of 112 pixels). Predictions for these patches are subsequently aggregated to produce the final segmentation maps.

This extended benchmark comprises a total of **87 distinct test scenarios**, encompassing synthetic, real, and rendered domain shifts. It provides a rigorous and comprehensive evaluation protocol that captures variations in resolution, scene diversity, object size, and semantic granularity.

# E  Computational Complexity

In this section, we provide an exhaustive analysis of each test-time adaptation method's resource footprint by measuring: (i) latency, (ii) floating-point operations (FLOPs), (iii) peak GPU memory usage, and (iv) number of learnable parameters. For a fair comparison, all measurements were performed on a single test sample using the same NVIDIA V100 (32 GB) GPU. The results are summarized in Table 8.

We compare TENT, TPT, CLIPArTT, WATT, and our MLMP across all four complexity metrics. In TENT, WATT, CLIPArTT, and MLMP, only the LayerNorm parameters of the vision encoder are updated during adaptation, whereas TPT introduces additional learnable tokens at the input of the text encoder. Additionally TENT, WATT, and MLMP use a fixed sets of text features without a need for recalculating them during adaptation/evaluation, so all prompt templates can be encoded once (✓ in the "One-time Text Encoder" column) and then reused throughout both adaptation and evaluation phases. In contrast, TPT and CLIPArTT modify prompt embeddings or refine text templates during adaptation, requiring multiple forward passes through the text encoder.

In terms of latency, MLMP completes both adaptation and evaluation in just 0.582 ms, which is only marginally slower than TENT (0.480 ms) but substantially faster than WATT (5.215 ms) and CLIPArTT (3.525 ms). Despite leveraging multi-level fusion and multiple prompt templates, MLMP maintains a lightweight computational profile with only 82.4 GFLOPs and 82.9 GFLOPs for adaptation and evaluation, respectively—comparable to TENT and significantly lower than all other baselines. Furthermore, MLMP's peak memory usage is among the lowest at 2,093.9 MB, and like TENT, it updates only 0.02% of the model parameters. These results highlight MLMP's efficiency: it delivers rich representational capacity through multi-level and multi-prompt integration by boosting the results significantly (as shown in Table 5 of the main paper), yet remains almost as lightweight as the simplest baseline.

# F  Performance with a Single Test Sample

As shown in Table 9, while TENT yields improvements on the original dataset, it leads to a performance drop on V20-C. In contrast, our method, MLMP, consistently improves performance over the

Table 8: Computational-complexity comparison across methods. For GFLOPs, only the forward-pass cost in adaptation and evaluation is measured; by common practice, the cost of back-propagation in adaptation phase can be approximated as twice the forward cost. A ✓ in the second column indicates that all text information are encoded *once* and cached. A dagger (†) indicates that TPT adds additional parameters beyond the original network.

| Method | One-time Text Encoder | Time (sec.) ↓ | | GFLOPs ↓ | | Max Memory (MB) ↓ | Learn. Params (Ratio) ↓ |
|---|---|---|---|---|---|---|---|
| | | Adapt | Eval | Adapt | Eval | | |
| TENT | ✓ | 0.462 | 0.018 | 79.1 | 79.1 | 2,068.4 | 102,400 (0.02%) |
| TPT | ✗ | 0.445 | 0.031 | 275.6 | 275.6 | 2,583.1 | 3,072† (<0.01%) |
| CLIPArTT | ✗ | 3.494 | 0.031 | 1,755.5 | 275.6 | 8,928.5 | 102,400 (0.02%) |
| WATT | ✓ | 5.197 | 0.018 | 553.9 | 79.1 | 7,232.4 | 716,800 (0.17%) |
| **MLMP (ours)** | ✓ | 0.541 | 0.041 | 82.4 | 82.9 | 2,093.9 | 102,400 (0.02%) |

no adaptation baseline, with gains of 8.77% on the otiginal data and 9.40% on the average across corruptions.

Table 9: mIoU performance comparison when using a single test sample for V20 dataset.

| OVSS Backbone: NACLIP | Adaptation Method | | |
|---|---|---|---|
| Dataset: V20 | No Adapt. | TENT | MLMP |
| Original | 75.91 | 76.20 | **84.68** |
| Gaussian noise | 62.89 | 62.59 | **71.92** |
| Shot noise | 66.26 | 65.45 | **75.78** |
| Impulse noise | 63.16 | 63.34 | **72.35** |
| Defocus blur | 72.59 | 71.37 | **80.91** |
| Glass blur | 71.44 | 69.95 | **79.29** |
| Motion blur | 73.10 | 72.55 | **81.64** |
| Zoom blur | 59.03 | 60.64 | **69.99** |
| Snow | 71.49 | 70.03 | **80.64** |
| Frost | 65.38 | 65.96 | **74.33** |
| Fog | 70.69 | 69.39 | **80.77** |
| Brightness | 74.95 | 74.73 | **84.58** |
| Contrast | 71.51 | 69.74 | **79.78** |
| Elastic transform | 62.86 | 65.09 | **75.32** |
| Pixelate | 77.28 | 75.84 | **85.63** |
| JPEG compression | 72.59 | 70.85 | **83.20** |
| V20-C Average | 69.01 | 68.50 | **78.41** |

# G  Generalization Across Model Variants

To evaluate the robustness and generality of our method, we conduct a series of experiments using different model configurations within the segmentation pipeline. Specifically, we assess how MLMP performs when (i) changing the vision transformer backbone, (ii) switching between different open-vocabulary semantic segmentation (OVSS) formulations, and (iii) adopting an entirely different vision–language model. These experiments demonstrate that our method maintains consistent improvements across a wide range of architectural and algorithmic configurations, confirming its flexibility and transferability.

## G.1  Comparison Across ViT Backbones

To evaluate whether our method generalizes across different vision transformer backbones, we replicate the main experiments using ViT-B/16 and ViT-B/32 in place of the default ViT-L/14 model. These backbones represent lighter configurations, with fewer parameters and larger patch (32 and 16). As shown in Tables 10 and 11, MLMP continues to provide substantial improvements over all

baselines across both configurations, on both clean data (V20 Original) and under severe synthetic corruptions. These results confirm that the benefits of our multi-level and multi-prompt adaptation strategy are not tied to model scale or specific architectural configurations, and remain effective even in lower-resolution settings.

Table 10: Performance comparison of test-time adaptation methods using the ViT-B/16 backbone with NaCLIP as the OVSS model. Results are reported as mIoU scores on the V20 dataset (original) and 15 corruption types. MLMP achieves the highest performance across all settings, demonstrating strong robustness even with a smaller backbone.

| OVSS Backbone: NACLIP | Adaptation Method | | | | | |
|---|---|---|---|---|---|---|
| Dataset: V20 | No Adapt. | TENT | TPT | WATT | CLIPArTT | MLMP |
| Original | 77.62 | 79.15 ±0.06 | 77.63 ±0.01 | 43.18 ±0.07 | 72.70 ±0.17 | **84.18** ±**0.07** |
| Gaussian noise | 48.00 | 52.04 ±0.12 | 48.11 ±0.00 | 18.83 ±0.14 | 33.80 ±0.52 | **61.67** ±**0.00** |
| Shot noise | 52.49 | 55.56 ±0.16 | 52.41 ±0.00 | 19.86 ±0.12 | 37.62 ±0.33 | **64.97** ±**0.10** |
| Impulse noise | 49.51 | 52.87 ±0.11 | 49.41 ±0.01 | 19.12 ±0.23 | 35.92 ±0.05 | **61.15** ±**0.09** |
| Defocus blur | 68.03 | 69.85 ±0.04 | 67.88 ±0.01 | 37.54 ±0.17 | 56.77 ±0.19 | **76.71** ±**0.02** |
| Glass blur | 62.17 | 65.14 ±0.17 | 62.45 ±0.00 | 31.12 ±0.08 | 47.86 ±0.15 | **72.62** ±**0.04** |
| Motion blur | 69.56 | 71.93 ±0.02 | 69.54 ±0.01 | 36.89 ±0.15 | 58.64 ±0.29 | **77.08** ±**0.03** |
| Zoom blur | 47.34 | 52.30 ±0.19 | 47.38 ±0.01 | 22.83 ±0.12 | 33.52 ±0.20 | **59.05** ±**0.20** |
| Snow | 60.88 | 64.38 ±0.05 | 61.24 ±0.00 | 27.68 ±0.21 | 49.91 ±0.38 | **71.41** ±**0.15** |
| Frost | 55.45 | 58.38 ±0.14 | 55.44 ±0.00 | 29.66 ±0.23 | 46.94 ±0.04 | **67.42** ±**0.09** |
| Fog | 67.07 | 70.01 ±0.02 | 67.07 ±0.00 | 35.84 ±0.23 | 59.98 ±0.09 | **76.32** ±**0.02** |
| Brightness | 73.33 | 75.14 ±0.17 | 73.23 ±0.00 | 40.87 ±0.09 | 67.30 ±0.33 | **82.14** ±**0.03** |
| Contrast | 60.30 | 63.30 ±0.04 | 60.20 ±0.00 | 29.68 ±0.12 | 48.42 ±0.55 | **70.02** ±**0.04** |
| Elastic transform | 50.14 | 54.83 ±0.01 | 50.00 ±0.01 | 23.32 ±0.14 | 43.67 ±0.00 | **63.65** ±**0.07** |
| Pixelate | 75.48 | 77.44 ±0.03 | 75.31 ±0.01 | 42.26 ±0.08 | 67.33 ±0.08 | **83.29** ±**0.01** |
| JPEG compression | 69.17 | 70.97 ±0.07 | 69.15 ±0.01 | 34.94 ±0.07 | 60.30 ±0.54 | **79.35** ±**0.12** |
| V20-C Average | 60.59 | 63.61 | 60.59 | 30.03 | 49.87 | **71.12** |

Table 11: Performance comparison of test-time adaptation methods using the ViT-B/32 backbone with NaCLIP as the OVSS model. Results are reported as mIoU scores on the V20 dataset (original) and 15 corruption types. MLMP achieves the highest performance across all settings, demonstrating strong robustness even with a smaller backbone and smaller patch size.

| OVSS Backbone: NACLIP | Adaptation Method | | | | | |
|---|---|---|---|---|---|---|
| Dataset: V20 | No Adapt. | TENT | TPT | WATT | CLIPArTT | MLMP |
| Original | 72.43 | 72.83 ±0.01 | 72.52 ±0.00 | 50.71 ±0.10 | 67.67 ±0.15 | **79.95** ±**0.01** |
| Gaussian noise | 47.59 | 49.30 ±0.18 | 47.43 ±0.00 | 29.26 ±0.09 | 37.36 ±0.17 | **59.27** ±**0.16** |
| Shot noise | 51.80 | 54.43 ±0.21 | 51.83 ±0.00 | 31.82 ±0.15 | 41.07 ±0.21 | **63.73** ±**0.12** |
| Impulse noise | 48.79 | 51.59 ±0.11 | 48.79 ±0.00 | 30.31 ±0.04 | 37.65 ±0.17 | **58.81** ±**0.04** |
| Defocus blur | 60.23 | 61.86 ±0.07 | 60.17 ±0.00 | 36.54 ±0.20 | 48.97 ±0.21 | **66.70** ±**0.08** |
| Glass blur | 54.59 | 56.73 ±0.09 | 54.80 ±0.00 | 28.70 ±0.10 | 37.96 ±0.26 | **65.34** ±**0.03** |
| Motion blur | 59.53 | 60.49 ±0.02 | 59.65 ±0.00 | 35.80 ±0.06 | 47.61 ±0.52 | **66.71** ±**0.08** |
| Zoom blur | 38.66 | 41.07 ±0.10 | 38.62 ±0.00 | 21.72 ±0.14 | 24.39 ±0.18 | **49.53** ±**0.08** |
| Snow | 49.17 | 51.01 ±0.08 | 48.88 ±0.01 | 27.55 ±0.17 | 40.30 ±0.00 | **62.30** ±**0.09** |
| Frost | 47.60 | 50.18 ±0.09 | 47.53 ±0.00 | 28.48 ±0.15 | 39.23 ±0.12 | **60.22** ±**0.07** |
| Fog | 56.25 | 59.47 ±0.08 | 56.22 ±0.00 | 35.76 ±0.09 | 47.63 ±0.29 | **68.18** ±**0.04** |
| Brightness | 68.03 | 68.94 ±0.04 | 67.95 ±0.00 | 46.11 ±0.02 | 61.07 ±0.25 | **76.90** ±**0.09** |
| Contrast | 48.79 | 50.67 ±0.07 | 48.88 ±0.02 | 29.54 ±0.06 | 36.64 ±0.05 | **58.81** ±**0.11** |
| Elastic transform | 52.73 | 55.59 ±0.20 | 52.75 ±0.00 | 29.86 ±0.21 | 44.84 ±0.18 | **65.39** ±**0.09** |
| Pixelate | 68.61 | 69.15 ±0.01 | 68.65 ±0.01 | 44.57 ±0.17 | 60.30 ±0.41 | **77.28** ±**0.04** |
| JPEG compression | 63.86 | 65.12 ±0.05 | 63.87 ±0.00 | 42.85 ±0.11 | 55.44 ±0.08 | **74.40** ±**0.09** |
| V20-C Average | 54.42 | 56.37 | 54.40 | 33.26 | 44.03 | **64.97** |

Table 12: Performance comparison of test-time adaptation methods using the original CLIP [1] and SCLIP [27] as the OVSS model with a ViT-B/16 backbone. MLMP consistently improves performance across all settings, highlighting its strong generalization capabilities.

| OVSS: **CLIP** [1] | Adaptation Method | | | OVSS: **SCLIP** [27] | Adaptation Method | | |
|---|---|---|---|---|---|---|---|
| Dataset: V20 | No Adapt. | TENT | MLMP | Dataset: V20 | No Adapt. | TENT | MLMP |
| Original | 33.11 | 51.36 ±0.00 | **61.47** ±0.01 | Original | 78.20 | 79.12 ±0.05 | **84.91** ±0.01 |
| Gaussian noise | 22.74 | 37.78 ±0.11 | **49.35** ±0.02 | Gaussian noise | 45.65 | 49.72 ±0.07 | **61.20** ±0.09 |
| Shot noise | 23.67 | 38.86 ±0.20 | **50.81** ±0.05 | Shot noise | 50.21 | 54.09 ±0.15 | **64.06** ±0.09 |
| Impulse noise | 22.36 | 36.63 ±0.15 | **47.81** ±0.17 | Impulse noise | 47.05 | 50.62 ±0.05 | **59.73** ±0.14 |
| Defocus blur | 31.83 | 48.33 ±0.13 | **58.25** ±0.04 | Defocus blur | 66.40 | 66.44 ±0.05 | **75.03** ±0.07 |
| Glass blur | 28.60 | 46.59 ±0.29 | **56.02** ±0.02 | Glass blur | 62.01 | 64.79 ±0.22 | **72.22** ±0.27 |
| Motion blur | 32.62 | 50.61 ±0.07 | **59.55** ±0.31 | Motion blur | 68.95 | 70.81 ±0.02 | **76.23** ±0.10 |
| Zoom blur | 23.42 | 39.43 ±0.06 | **47.65** ±0.10 | Zoom blur | 45.12 | 48.50 ±0.11 | **57.84** ±0.16 |
| Snow | 28.38 | 48.05 ±0.04 | **55.89** ±0.04 | Snow | 60.61 | 64.60 ±0.20 | **71.70** ±0.06 |
| Frost | 26.20 | 45.33 ±0.04 | **53.56** ±0.17 | Frost | 56.14 | 58.43 ±0.03 | **68.57** ±0.02 |
| Fog | 28.66 | 46.98 ±0.05 | **56.62** ±0.02 | Fog | 68.34 | 70.03 ±0.17 | **76.57** ±0.06 |
| Brightness | 33.71 | 51.77 ±0.24 | **60.96** ±0.08 | Brightness | 74.03 | 74.62 ±0.11 | **82.91** ±0.04 |
| Contrast | 26.06 | 42.86 ±0.08 | **53.55** ±0.04 | Contrast | 58.98 | 61.69 ±0.08 | **69.57** ±0.10 |
| Elastic transform | 27.12 | 45.92 ±0.11 | **51.02** ±0.21 | Elastic transform | 52.29 | 56.45 ±0.02 | **65.05** ±0.14 |
| Pixelate | 33.32 | 51.11 ±0.00 | **61.22** ±0.09 | Pixelate | 75.56 | 76.66 ±0.09 | **82.99** ±0.00 |
| JPEG compression | 31.64 | 49.76 ±0.04 | **59.79** ±0.06 | JPEG compression | 68.38 | 69.85 ±0.12 | **78.69** ±0.17 |
| V21-C Average | 28.02 | 45.33 | **54.80** | V21-C Average | 59.98 | 62.49 | **70.82** |

## G.2 Comparison Across OVSS Methods

Our method is designed to be flexible and agnostic to the underlying open-vocabulary semantic segmentation (OVSS) formulation. While all main experiments in the paper use NaCLIP as the OVSS baseline—paired with No Adapt., TENT, TPT, CLIPArTT, WATT, and our MLMP—we further evaluate the generality of our approach by applying MLMP to two alternative OVSS methods.

Specifically, we consider the original CLIP [1], adapted for pixel-wise segmentation via patch-level similarity, and SCLIP [27], which incorporates spatial priors into the vision-language matching process. As shown in Table 12, applying MLMP on top of these OVSS baselines yields consistent improvements across both clean and corrupted settings.

## G.3 Generalization to Emerging VLMs

To further assess the generality of our approach, we evaluate MLMP on SigLIP v2 [47], one of the most recently introduced vision–language models. As shown in Table 13, MLMP continues to deliver consistent improvements across both natural and corrupted datasets, indicating that the core components of our method—multi-level and multi-prompt aggregation—generalize well beyond CLIP-based architectures.

# H Effect of Longer Prompts

Recent studies such as Long-CLIP [48] and TULIP [49] have explored extending the text length capability of vision–language models, suggesting that richer linguistic descriptions may improve alignment. Motivated by this, we investigated whether incorporating longer and more descriptive prompts could further enhance MLMP's performance.

To this end, we generated extended templates derived from our original seven class-agnostic prompts using ChatGPT, together with extended class names where each category was expressed in full natural language. For example, "a photo of the large {class}" becomes "a photograph of a very large {class}, where the size of the object dominates the frame or is shown in contrast to smaller elements," while "aeroplane" becomes "a powered flying vehicle with fixed wings and engines, designed to transport people or cargo through the air over long distances."

As summarized in Table 14, all experiments were conducted using the Long-CLIP [48] backbone. We evaluate three text variants: (1) the default short templates used throughout our main experiments, (2) *Extended Templates*, where each template is replaced with a longer and more descriptive sentence, and (3) *Extended Classes*, where category names are expanded into full natural-language descriptions.

Table 13: Performance comparison of test-time adaptation using SigLIP-2 as the OVSS model. Results are reported as mIoU scores on the V20 dataset. MLMP shows improvement over the baseline in most corruption types.

| OVSS Backbone: SigLIP-2 | Adaptation Method | |
| --- | --- | --- |
| Dataset: V20 | No Adapt. | MLMP |
| Original | 66.62 | **67.88** ±0.32 |
| Gaussian noise | 39.74 | **41.05** ±0.34 |
| Shot noise | 40.24 | **43.56** ±0.07 |
| Impulse noise | 40.76 | **42.43** ±0.22 |
| Defocus blur | 47.15 | **50.41** ±0.02 |
| Glass blur | 38.64 | **40.60** ±0.55 |
| Motion blur | 42.30 | **42.98** ±0.39 |
| Zoom blur | 27.29 | **29.20** ±0.03 |
| Snow | 3.85 | **4.84** ±0.03 |
| Frost | 21.32 | **23.12** ±0.02 |
| Fog | 70.73 | **70.86** ±0.04 |
| Brightness | 18.45 | **19.14** ±0.27 |
| Contrast | 70.88 | **70.35** ±0.06 |
| Elastic transform | 38.41 | **38.71** ±0.05 |
| Pixelate | 57.85 | **59.14** ±0.07 |
| JPEG compression | 56.35 | **58.55** ±0.04 |
| V20-C Average | 40.93 | **42.33** |

Table 14: Effect of longer prompts and extended class descriptions on segmentation performance (mIoU). All experiments use Long-CLIP as the OVSS model. Each pair of columns shows results without and with MLMP adaptation.

| OVSS: Long-CLIP | Adaptation Setting | | | | | |
| --- | --- | --- | --- | --- | --- | --- |
| Dataset | Default Templates | | Extended Templates | | Extended Classes | |
| | No Adapt. | + MLMP | No Adapt. | + MLMP | No Adapt. | + MLMP |
| V20 | 39.64 | 54.45 ±0.09 | 35.55 | 53.43 ±0.06 | 25.66 | 43.92 ±0.04 |
| V20-C Average | 29.36 | 48.71 | 26.92 | 48.90 | 21.09 | 39.01 |
| V21 | 16.83 | 19.54 ±0.02 | 15.87 | 19.15 ±0.04 | 11.27 | 15.25 ±0.03 |
| V21-C Average | 13.64 | 18.57 | 12.98 | 18.27 | 10.23 | 14.66 |

For each variant, we report results with and without our MLMP adaptation (+ *MLMP*). All scores correspond to mIoU.

Overall, these results indicate that MLMP's adaptation mechanism effectively handles both concise and extended textual inputs without overfitting to prompt verbosity. We include this analysis to provide a clearer picture of MLMP's behavior under longer or more descriptive prompts.

# I  Effect of Adaptation Iterations

The number of adaptation iterations in TTA varies across the literature. While some studies evaluate performance under a single adaptation step, others perform multiple iterations to improve convergence. In our main experiments, we followed the common 10-iteration setup adopted in prior TTA works [8, 5, 37], as it provides a consistent protocol for comparing vision–language adaptation methods such as CLIPArTT and WATT. For fairness, we used identical hyperparameters across all methods.

To further examine the influence of the iteration count, we also evaluated MLMP under a stricter *one-iteration* protocol, where only a single adaptation step is performed. As summarized in Table 15, MLMP still improves over the baseline and outperforms TENT even with just one iteration, which demonstrates the effectiveness of our method even under a stricter setting.

Table 15: Evaluation of MLMP under a single-iteration TTA protocol compared to standard baselines (mIoU). All experiments use NACLIP as the OVSS model.

| OVSS: NACLIP | Adaptation Method | | |
|---|---|---|---|
| Dataset | No Adapt. | TENT | MLMP |
| V20 | 75.91 | 76.80 $\pm0.01$ | **79.88** $\pm0.02$ |
| V20-C Average | 69.01 | 70.15 | **73.91** |
| V21 | 45.12 | 45.36 $\pm0.00$ | **50.24** $\pm0.01$ |
| V21-C Average | 40.75 | 41.02 | **46.07** |

Table 16: Comparison of episodic (reset-based) and online adaptation settings for MLMP and TENT (mIoU). All experiments use NACLIP as the OVSS model.

| OVSS: NACLIP | Adaptation Setting | | | | |
|---|---|---|---|---|---|
| Dataset | No Adapt. | Episodic: TENT | Episodic: MLMP | Online: TENT | Online: MLMP |
| V20 | 75.91 | 77.00 $\pm0.04$ | **83.76** $\pm0.00$ | 76.44 $\pm0.21$ | 79.19 $\pm0.45$ |
| V20-C Average | 69.01 | 69.33 | **77.58** | 69.32 | 71.77 |

## J  Episodic vs. Online Adaptation

We further investigate the difference between *episodic* (reset-based) and *online* adaptation settings in test-time adaptation (TTA). In our main experiments, all methods—including MLMP and baselines—were evaluated under the episodic setup, where the model is reset to its initial weights after each batch. This protocol avoids cumulative error propagation and is widely used in prior TTA works [8, 37, 5].

To analyze the impact of continuous updates, we also tested an *online* version of MLMP, where model parameters are updated sequentially without resets and carried over to the next batch. We used identical hyperparameters as in the episodic setting, and further experimented with lower learning rates and fewer adaptation steps for stability.

As shown in Table 16, we observe that while the online variant improves over baselines, it still underperforms compared to the reset-based (episodic) setting. We believe this is primarily due to error accumulation and model drift, as documented in prior work showing that some recent classification TTA methods exhibit performance degradation in an online setting—even when the distribution is static [50]. This issue is particularly pronounced in semantic segmentation, where dense, spatial predictions make the model more sensitive to incorrect updates, and entropy minimization can amplify early misclassifications. Over time, this leads the model to drift away from its well-initialized source parameters, reducing its ability to generalize across the target domain.

Episodic (reset) adaptation avoids the runaway effects of carrying over errors, which is why it tends to be safer [2]. In contrast, fully online adaptation—though attractive for its potential to continuously refine the model—must confront these challenges.

We believe that online adaptation in segmentation can be improved through several future directions: (1) introducing occasional resets or using exponential moving average (EMA) of model weights to limit drift and reduce accumulated errors; (2) filtering high-entropy samples or gradients, which is particularly important in segmentation due to its spatial granularity and sensitivity to local noise; and (3) incorporating an auxiliary self-supervised objective, such as rotation prediction [1], to provide a more reliable adaptation signal—though this may come with increased computational cost. Notably, the fact that our method works effectively with very few test samples and does not rely on accumulating state makes it efficient for real-world applications.

## K  Visualization of Layer-Wise Confidence Weights

In the main paper (Figure 3), we presented the layer-wise confidence weights for the V20, Cityscapes, and COCO-Stuff datasets. Here, we extend the analysis by visualizing the weights for four additional datasets: V21, P59, P60, and COCO-Object in Figure 5. As with the earlier results, we plot the mean and standard deviation of the learned weights across layers under various corruption types.

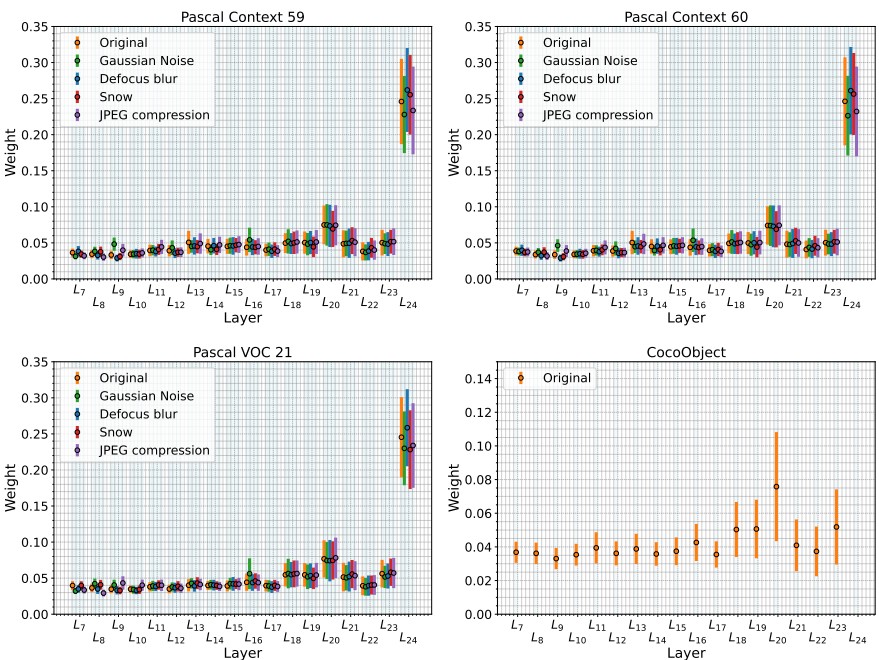

Figure 5: Mean and standard deviation of weights of intermediate layers for several datasets.

The observed trends closely mirror those reported in the main paper. In particular, deeper layers consistently receive higher confidence scores, while lower and mid-level layers still contribute under corrupted conditions. This reweighting behavior reflects the adaptive nature of our fusion strategy, which dynamically emphasizes the most reliable representations depending on the dataset and corruption type. These results further support the robustness and generality of our layer-wise uncertainty-aware fusion mechanism across diverse segmentation benchmarks.

## L    Visualization of Segmentation Maps

Figure 6 presents qualitative segmentation results on the v20 dataset, comparing predictions from the non-adaptive baseline, TENT, and our MLMP method. MLMP demonstrates stronger spatial consistency and fewer semantic errors, particularly in challenging regions where both the baseline and TENT struggle. The integration of intermediate-layer supervision appears especially beneficial for refining small object boundaries and correcting fine-grained details.

In Figures 7–15, we provide additional qualitative examples across a range of corruption types, using NaCLIP as the our OVSS backbone. These include both successful and failure cases under Gaussian noise, defocus blur, snow, and JPEG compression. Overall, the results illustrate the robustness of MLMP in mitigating noise-induced artifacts and improving prediction confidence, while also highlighting failure modes that warrant further investigation.

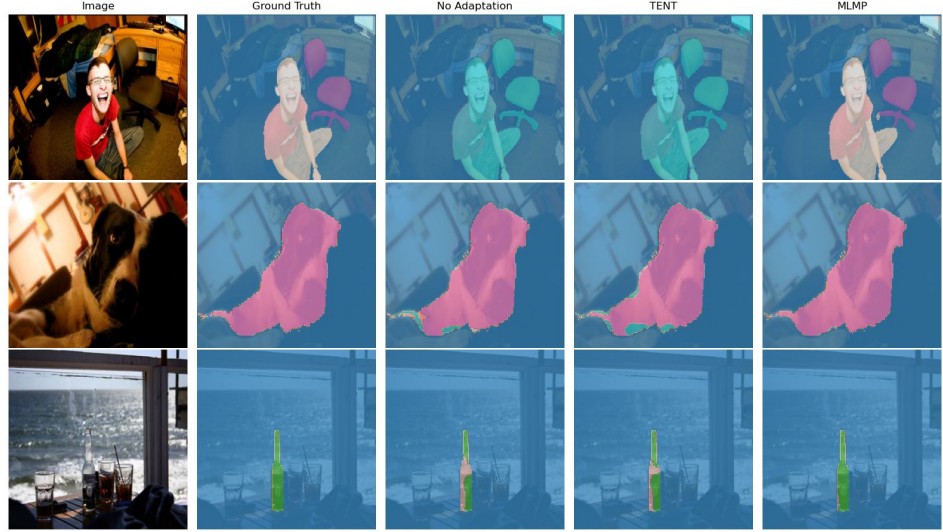

Figure 6: Qualitative results for No Adapt., TENT and MLMP on V20 Original.

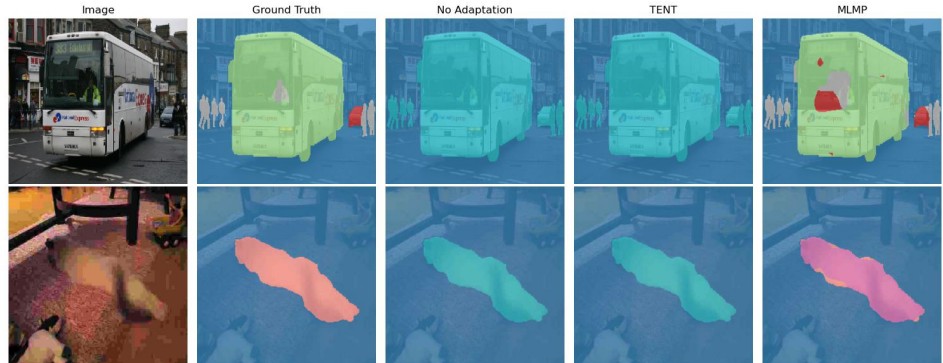

Figure 7: Failed Cases of MLMP on V20 Original.

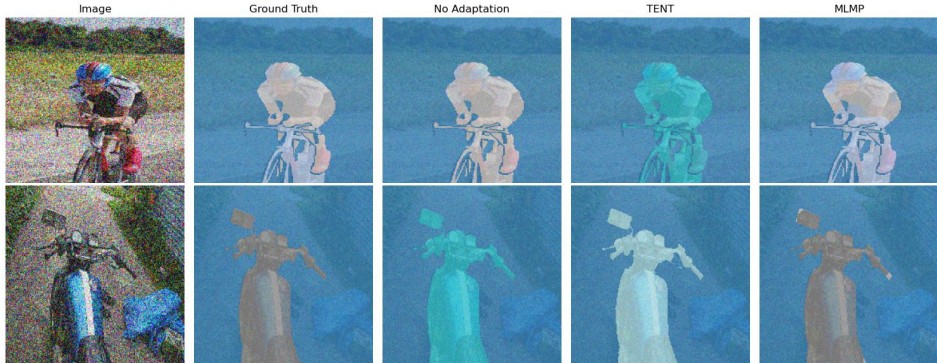

Figure 8: Good Cases of MLMP on V20 for V20 gaussian noise corruption.

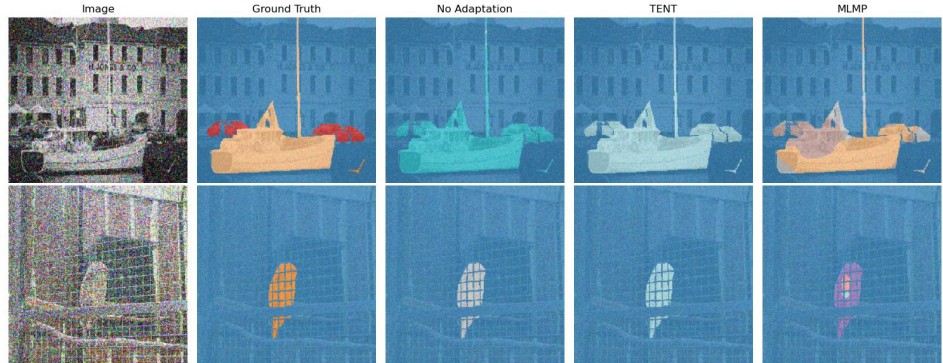

Figure 9: Failed Cases of MLMP on V20 gaussian noise corruption.

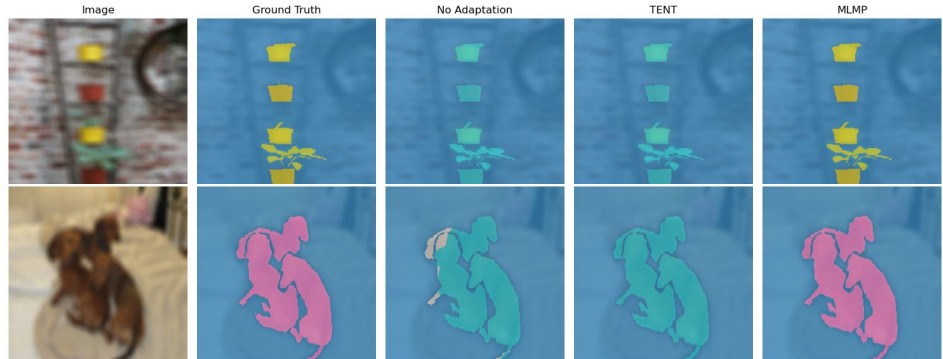

Figure 10: Good Cases of MLMP on V20 defocus blur corruption.

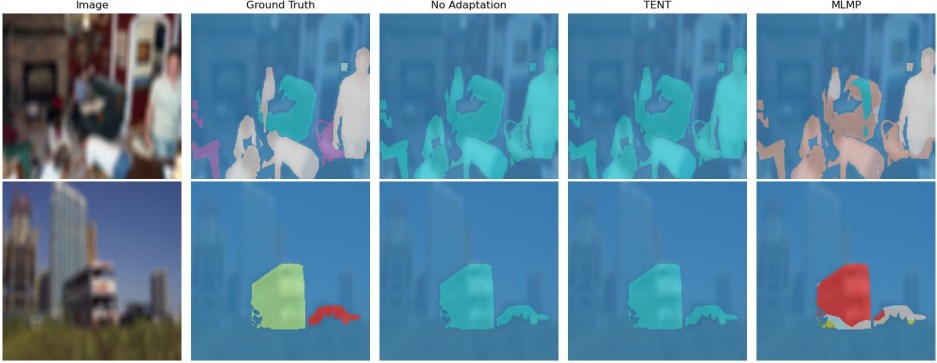

Figure 11: Failed Cases of MLMP on V20 defocus blur corruption.

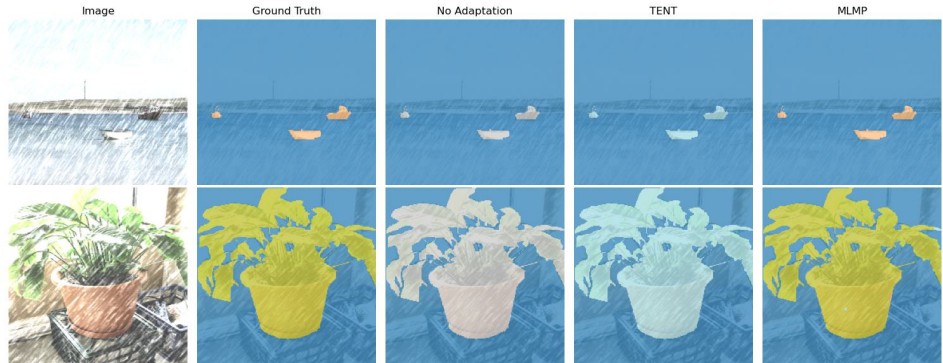
Figure 12: Good Cases of MLMP on V20 snow corruption.

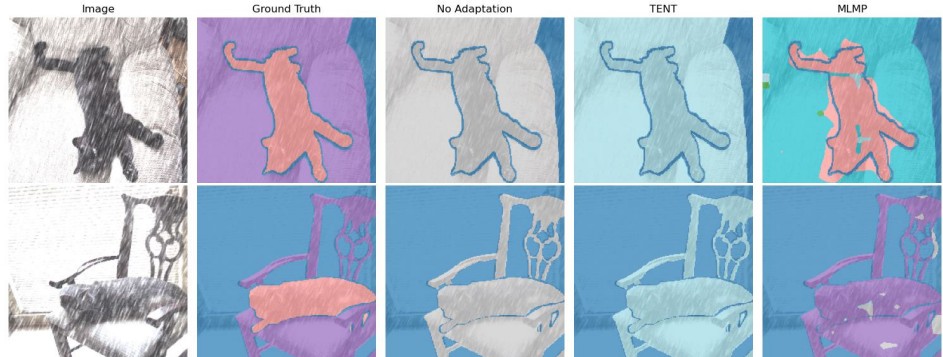
Figure 13: Failed Cases of MLMP on V20 snow corruption.

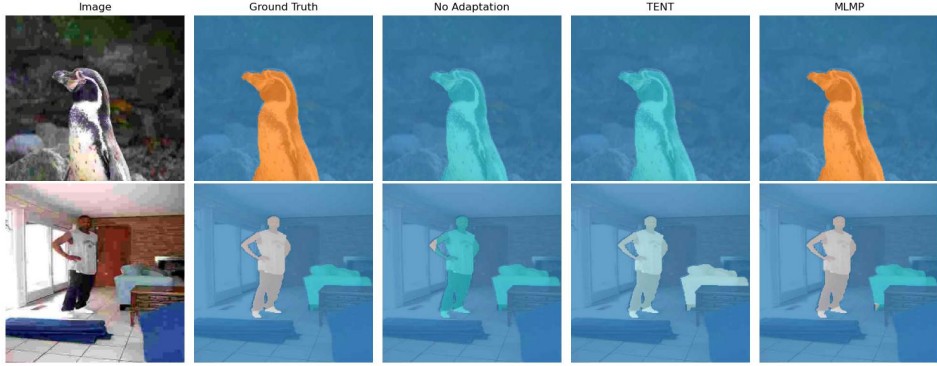
Figure 14: Good Cases of MLMP on V20 JPEG compression corruption.

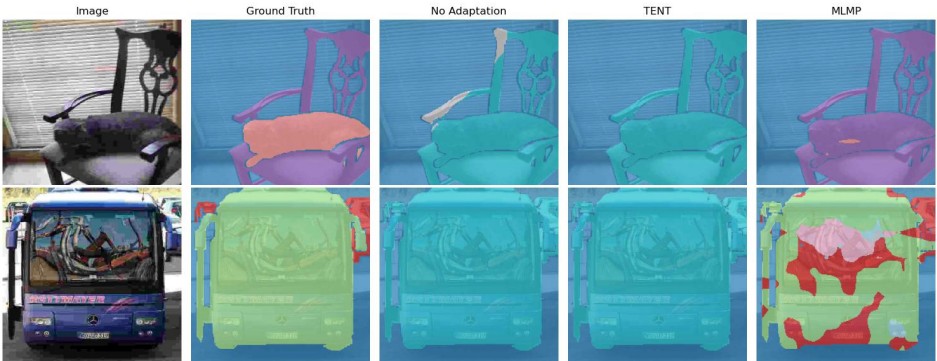

Figure 15: Failed Cases of MLMP on V20 JPEG compression corruption.

## M    Detailed Results of the Main Paper

This section provides complete versions of the experimental results that were summarized or partially reported in the main paper. It includes full tables for ablation studies and detailed comparisons with baseline adaptation methods across various datasets and evaluation settings. These results offer a more comprehensive view of the effectiveness and robustness of our proposed approach.

### M.1    Ablation Studies

We provide here the detailed versions of the tables referenced in the ablation section of the main paper. Table 17 shows the detailed results of using different layer ranges in the proposed multi-level adaptation. Table 18 reports results for different strategies to integrate different prompt templates. Table 19 presents a detailed analysis of the impact of the number of templates. Finally, Table 20 presents results for the combination of all proposed components, showing individual and combined contributions.

### M.2    Comparison with Alternative Adaptation Methods

This section presents comprehensive tables with the full experimental results corresponding to those summarized in the main paper. Table 21 reports detailed results for the V21 dataset, Table 22 for P59, Table 23 for P60, and Table 24 for the Cityscapes dataset. In addition to the datasets reported above, we also include detailed results on the **ACDC** [45] dataset, which is specifically designed to capture real-world adverse conditions such as fog, night, rain, and snow. It is worth mentioning that in ACDC, approximately half of the adverse-condition images have corresponding *reference (clean)* views captured at nearly the same locations. While these reference images are captured at approximately the same location as the shifted ones, the scene content may differ (e.g., different vehicles or objects present) due to real-world variability. Following our main protocol, we perform test-time adaptation independently on both reference and adverse views, and report the mIoU for each condition in Table 25. The results show that MLMP consistently improves over TENT and the non-adapted baseline across all conditions, confirming its robustness to real, non-synthetic distribution shifts.

Table 17: mIoU performance over different layer aggregation strategies. Maximum value in each row is highlighted.

| ViT-L/14 Layer Range | $L_{24}$ (last) | $L_{23\text{-}24}$ (last two) | $L_{22\text{-}24}$ (last three) | $L_{19\text{-}24}$ (last 25%) | $L_{13\text{-}24}$ (last 50%) | $L_{7\text{-}24}$ (last 75%) | $L_{1\text{-}24}$ (all layers) |
|---|---|---|---|---|---|---|---|
| Original (V20) | 77.00 ±0.04 | 77.65 ±0.02 | 77.66 ±0.09 | 80.61 ±0.05 | 80.50 ±0.03 | **81.67 ±0.04** | 78.79 ±0.02 |
| Gaussian noise | 63.02 ±0.06 | 64.41 ±0.05 | 65.39 ±0.13 | 66.88 ±0.18 | 66.88 ±0.02 | **67.82 ±0.01** | 63.06 ±0.09 |
| Shot noise | 65.88 ±0.06 | 67.74 ±0.11 | 68.43 ±0.04 | 70.11 ±0.09 | 70.40 ±0.02 | **70.52 ±0.12** | 64.88 ±0.02 |
| Impulse noise | 64.17 ±0.04 | 65.53 ±0.09 | 66.19 ±0.12 | 67.24 ±0.17 | 67.15 ±0.09 | **68.10 ±0.01** | 62.09 ±0.05 |
| Defocus blur | 72.06 ±0.12 | 72.93 ±0.19 | 72.84 ±0.02 | 76.10 ±0.16 | 76.37 ±0.05 | **78.78 ±0.02** | 77.56 ±0.09 |
| Glass blur | 70.74 ±0.07 | 71.73 ±0.06 | 72.53 ±0.15 | 74.20 ±0.12 | 75.53 ±0.05 | **77.66 ±0.05** | 75.85 ±0.02 |
| Motion blur | 73.50 ±0.10 | 73.74 ±0.08 | 74.62 ±0.09 | 76.83 ±0.07 | 77.50 ±0.07 | **79.39 ±0.16** | 77.22 ±0.15 |
| Zoom blur | 61.36 ±0.07 | 62.10 ±0.01 | 62.56 ±0.00 | 63.99 ±0.20 | 64.59 ±0.14 | **66.50 ±0.16** | 64.79 ±0.01 |
| Snow | 71.04 ±0.05 | 72.09 ±0.04 | 72.56 ±0.02 | 74.47 ±0.12 | 74.41 ±0.01 | **76.39 ±0.02** | 73.72 ±0.07 |
| Frost | 67.01 ±0.02 | 66.92 ±0.02 | 67.23 ±0.04 | 68.94 ±0.01 | 70.03 ±0.01 | **71.17 ±0.00** | 68.23 ±0.02 |
| Fog | 70.54 ±0.07 | 71.44 ±0.07 | 71.50 ±0.05 | 74.50 ±0.16 | 74.62 ±0.08 | **76.55 ±0.06** | 74.37 ±0.03 |
| Brightness | 75.61 ±0.02 | 76.27 ±0.00 | 76.80 ±0.04 | 79.16 ±0.07 | 79.64 ±0.16 | **81.34 ±0.04** | 79.11 ±0.05 |
| Contrast | 70.51 ±0.04 | 71.72 ±0.13 | 72.17 ±0.00 | 74.58 ±0.08 | 75.42 ±0.09 | **76.87 ±0.06** | 74.09 ±0.00 |
| Elastic transform | 65.78 ±0.05 | 66.08 ±0.08 | 66.06 ±0.06 | 68.62 ±0.15 | 70.38 ±0.09 | **70.59 ±0.14** | 67.91 ±0.12 |
| Pixelate | 76.95 ±0.12 | 78.38 ±0.02 | 78.46 ±0.05 | 80.70 ±0.05 | 81.11 ±0.03 | **83.02 ±0.04** | 81.53 ±0.02 |
| JPEG compression | 71.84 ±0.15 | 73.88 ±0.11 | 74.40 ±0.07 | 76.96 ±0.02 | 77.67 ±0.03 | **78.73 ±0.08** | 75.87 ±0.19 |
| V20-C Average | 69.33 | 70.33 | 70.78 | 72.89 | 73.45 | **74.90** | 72.02 |

Table 18: mIoU performance comparison for different strategies to integrate different prompt templates.

| Dataset: V20 | Text | Params | Loss |
|---|---|---|---|
| Original | 78.91 ±0.07 | 74.46 ±0.21 | **79.70 ±0.06** |
| Gaussian noise | 66.27 ±0.00 | 62.83 ±0.04 | **66.75 ±0.01** |
| Shot noise | 69.78 ±0.10 | 67.10 ±0.12 | **70.03 ±0.04** |
| Impulse noise | 66.73 ±0.03 | 64.57 ±0.52 | **67.88 ±0.07** |
| Defocus blur | 74.05 ±0.10 | 70.28 ±0.16 | **74.31 ±0.09** |
| Glass blur | 73.31 ±0.08 | 70.28 ±0.34 | **74.38 ±0.36** |
| Motion blur | 75.20 ±0.09 | 71.56 ±0.09 | **75.72 ±0.04** |
| Zoom blur | 63.31 ±0.11 | 61.29 ±0.09 | **64.61 ±0.01** |
| Snow | 73.78 ±0.02 | 70.10 ±0.30 | **74.66 ±0.01** |
| Frost | 68.90 ±0.06 | 66.42 ±0.04 | **69.50 ±0.01** |
| Fog | 72.60 ±0.03 | 67.63 ±0.07 | **73.33 ±0.03** |
| Brightness | 78.16 ±0.02 | 74.05 ±0.03 | **78.69 ±0.02** |
| Contrast | 73.75 ±0.06 | 69.47 ±0.24 | **74.09 ±0.01** |
| Elastic transform | 68.41 ±0.02 | 64.98 ±0.09 | **69.14 ±0.05** |
| Pixelate | 79.59 ±0.06 | 75.54 ±0.05 | **80.09 ±0.03** |
| JPEG compression | 74.98 ±0.05 | 70.55 ±0.11 | **75.56 ±0.02** |
| V20-C Average | 71.92 | 68.44 | **72.58** |

Table 19: IoU performance of our method for different numbers of templates.

| Dataset: V20 | 1 Template | 3 Templates | 7 Templates | 20 Templates | 80 Templates |
|---|---|---|---|---|---|
| Original | 77.00 ±0.04 | 79.48 ±0.08 | **79.70 ±0.06** | 79.17 ±0.01 | 79.25 ±0.02 |
| Gaussian noise | 63.02 ±0.06 | 66.51 ±0.10 | **66.75 ±0.01** | 65.94 ±0.09 | 66.02 ±0.02 |
| Shot noise | 65.88 ±0.06 | **70.20 ±0.03** | 70.03 ±0.04 | 68.59 ±0.01 | 69.00 ±0.01 |
| Impulse noise | 64.17 ±0.04 | 67.57 ±0.01 | **67.88 ±0.07** | 66.53 ±0.06 | 66.91 ±0.02 |
| Defocus blur | 72.06 ±0.12 | **74.66 ±0.02** | 74.31 ±0.09 | 73.89 ±0.12 | 74.09 ±0.06 |
| Glass blur | 70.74 ±0.07 | **74.38 ±0.12** | 74.38 ±0.36 | 73.78 ±0.09 | 73.88 ±0.07 |
| Motion blur | 73.50 ±0.10 | **75.80 ±0.08** | 75.72 ±0.04 | 75.19 ±0.04 | 75.19 ±0.04 |
| Zoom blur | 61.36 ±0.07 | 63.47 ±0.02 | **64.61 ±0.01** | 63.50 ±0.03 | 63.84 ±0.04 |
| Snow | 71.04 ±0.05 | 74.09 ±0.08 | **74.66 ±0.01** | 73.78 ±0.03 | 73.91 ±0.01 |
| Frost | 67.01 ±0.02 | 69.09 ±0.01 | **69.50 ±0.01** | 69.08 ±0.00 | 69.10 ±0.02 |
| Fog | 70.54 ±0.07 | **73.69 ±0.01** | 73.33 ±0.03 | 72.87 ±0.03 | 72.86 ±0.05 |
| Brightness | 75.61 ±0.02 | 78.49 ±0.05 | **78.69 ±0.02** | 77.86 ±0.02 | 78.06 ±0.02 |
| Contrast | 70.51 ±0.04 | 74.01 ±0.22 | **74.09 ±0.01** | 73.48 ±0.04 | 73.56 ±0.09 |
| Elastic transform | 65.78 ±0.05 | 68.09 ±0.01 | **69.14 ±0.05** | 67.94 ±0.02 | 68.31 ±0.09 |
| Pixelate | 76.95 ±0.12 | 79.91 ±0.04 | **80.09 ±0.03** | 79.15 ±0.06 | 79.41 ±0.05 |
| JPEG compression | 71.84 ±0.15 | 75.00 ±0.05 | **75.56 ±0.02** | 74.45 ±0.09 | 74.75 ±0.01 |
| V20-C Average | 69.33 | 72.33 | **72.58** | 71.74 | 71.93 |

Table 20: Detailed mIoU comparison of MLMP components, showing individual and combined contributions.

| Multi-Level Fusion | ✗ | ✓ | ✓ | ✗ | ✗ | ✓ | ✓ | ✓ | ✓ | ✗ | ✓ | ✓ |
|---|---|---|---|---|---|---|---|---|---|---|---|---|
| Multi-Prompt Loss | ✗ | ✗ | ✗ | ✓ | ✗ | ✓ | ✓ | ✗ | ✗ | ✓ | ✓ | ✓ |
| Image-Level Entropy | ✗ | ✗ | ✗ | ✗ | ✓ | ✗ | ✗ | ✓ | ✓ | ✓ | ✗ | ✓ |
| Uncertainty-Aware Weight. | ✗ | ✗ | ✓ | ✗ | ✗ | ✗ | ✓ | ✗ | ✓ | ✓ | ✗ | ✓ |
| Original | 77.00 ±0.04 | 77.38 ±0.01 | 81.67 ±0.04 | 79.70 ±0.06 | 78.74 ±0.08 | 78.97 ±0.03 | 83.00 ±0.03 | 77.69 ±0.01 | 82.70 ±0.01 | 81.15 ±0.02 | 79.13 ±0.00 | **83.76 ±0.00** |
| Gaussian noise | 63.02 ±0.06 | 65.42 ±0.04 | 67.82 ±0.01 | 66.75 ±0.01 | 65.66 ±0.04 | 65.96 ±0.04 | 69.13 ±0.07 | 66.17 ±0.04 | 69.00 ±0.05 | 69.62 ±0.01 | 67.35 ±0.09 | **71.13 ±0.09** |
| Shot noise | 65.88 ±0.06 | 67.49 ±0.01 | 70.52 ±0.12 | 70.03 ±0.04 | 68.97 ±0.03 | 69.05 ±0.03 | 72.31 ±0.01 | 69.50 ±0.06 | 73.22 ±0.03 | 72.89 ±0.02 | 71.02 ±0.05 | **75.02 ±0.03** |
| Impulse noise | 64.17 ±0.04 | 65.11 ±0.17 | 68.10 ±0.01 | 67.88 ±0.07 | 66.35 ±0.12 | 65.39 ±0.12 | 68.86 ±0.13 | 65.16 ±0.10 | 68.77 ±0.15 | 70.31 ±0.09 | 67.17 ±0.09 | **71.34 ±0.11** |
| Defocus blur | 72.06 ±0.12 | 76.65 ±0.23 | 78.78 ±0.02 | 74.31 ±0.09 | 75.00 ±0.07 | 76.46 ±0.15 | 78.78 ±0.10 | 77.29 ±0.10 | 79.78 ±0.05 | 77.14 ±0.05 | 77.79 ±0.19 | **80.36 ±0.06** |
| Glass blur | 70.74 ±0.07 | 75.24 ±0.05 | 77.66 ±0.05 | 74.38 ±0.36 | 73.54 ±0.23 | 75.46 ±0.09 | 77.48 ±0.01 | 75.71 ±0.10 | 78.09 ±0.04 | 76.17 ±0.08 | 76.87 ±0.07 | **78.84 ±0.05** |
| Motion blur | 73.50 ±0.10 | 77.16 ±0.13 | 79.39 ±0.16 | 75.72 ±0.04 | 76.09 ±0.09 | 77.72 ±0.09 | 79.97 ±0.04 | 78.81 ±0.03 | 81.49 ±0.02 | 78.25 ±0.05 | 79.00 ±0.07 | **81.41 ±0.05** |
| Zoom blur | 61.36 ±0.07 | 64.22 ±0.12 | 66.50 ±0.16 | 64.61 ±0.01 | 64.04 ±0.02 | 66.38 ±0.15 | 68.41 ±0.13 | 65.12 ±0.16 | 67.69 ±0.08 | 68.32 ±0.17 | 67.61 ±0.05 | **69.41 ±0.12** |
| Snow | 71.04 ±0.05 | 72.64 ±0.07 | 76.39 ±0.02 | 74.66 ±0.01 | 74.16 ±0.02 | 73.25 ±0.12 | 77.31 ±0.11 | 74.05 ±0.08 | 78.50 ±0.16 | 77.20 ±0.02 | 74.94 ±0.06 | **79.53 ±0.05** |
| Frost | 67.01 ±0.02 | 67.30 ±0.03 | 71.17 ±0.00 | 69.50 ±0.01 | 69.31 ±0.00 | 67.57 ±0.19 | 71.73 ±0.21 | 68.31 ±0.12 | 72.81 ±0.08 | 71.34 ±0.02 | 68.69 ±0.11 | **73.20 ±0.07** |
| Fog | 70.54 ±0.07 | 72.73 ±0.03 | 76.55 ±0.06 | 73.33 ±0.03 | 73.41 ±0.00 | 75.06 ±0.08 | 78.38 ±0.03 | 73.48 ±0.13 | 77.62 ±0.05 | 75.98 ±0.02 | 75.81 ±0.02 | **79.81 ±0.06** |
| Brightness | 75.61 ±0.02 | 77.23 ±0.07 | 81.34 ±0.04 | 78.69 ±0.02 | 77.34 ±0.01 | 77.69 ±0.02 | 82.00 ±0.03 | 77.20 ±0.05 | 82.16 ±0.03 | 80.63 ±0.05 | 78.27 ±0.06 | **83.51 ±0.01** |
| Contrast | 70.51 ±0.04 | 73.36 ±0.08 | 76.87 ±0.06 | 74.09 ±0.01 | 74.14 ±0.14 | 73.57 ±0.05 | 77.42 ±0.09 | 74.09 ±0.01 | 78.09 ±0.08 | 76.97 ±0.01 | 74.75 ±0.06 | **79.06 ±0.16** |
| Elastic transform | 65.78 ±0.05 | 65.38 ±0.12 | 70.59 ±0.14 | 69.14 ±0.05 | 67.92 ±0.06 | 68.76 ±0.04 | 72.96 ±0.02 | 66.78 ±0.03 | 71.80 ±0.02 | 71.53 ±0.03 | 69.77 ±0.07 | **74.03 ±0.01** |
| Pixelate | 76.95 ±0.12 | 79.53 ±0.03 | 83.02 ±0.04 | 80.09 ±0.03 | 79.09 ±0.03 | 80.77 ±0.05 | 83.93 ±0.06 | 79.85 ±0.08 | 83.90 ±0.00 | 81.92 ±0.12 | 81.34 ±0.07 | **84.97 ±0.04** |
| JPEG compression | 71.84 ±0.15 | 74.38 ±0.12 | 78.73 ±0.08 | 75.56 ±0.02 | 74.77 ±0.11 | 76.77 ±0.00 | 80.81 ±0.09 | 74.61 ±0.06 | 79.79 ±0.02 | 77.98 ±0.02 | 77.94 ±0.07 | **82.06 ±0.01** |
| V20-C Average | 69.33 | 71.59 | 74.90 | 72.58 | 71.99 | 72.66 | 75.97 | 72.41 | 76.18 | 75.08 | 73.89 | **77.58** |

Table 21: mIoU comparison of MLMP and baseline methods on the V21 dataset, evaluated on both the original images and 15 corruption types.

| OVSS Backbone: NACLIP | Adaptation Method | | | | | |
|---|---|---|---|---|---|---|
| Dataset: V21 | No Adapt. | TENT | TPT | WATT | CLIPArTT | MLMP |
| Original | 45.12 | 45.65 ±0.02 | 45.17 ±0.00 | 28.58 ±0.05 | 39.50 ±0.04 | **50.78 ±0.02** |
| Gaussian noise | 37.40 | 37.95 ±0.00 | 37.34 ±0.00 | 20.93 ±0.07 | 30.05 ±0.18 | **43.59 ±0.01** |
| Shot noise | 39.33 | 39.17 ±0.03 | 39.23 ±0.00 | 22.06 ±0.06 | 32.07 ±0.17 | **45.55 ±0.02** |
| Impulse noise | 37.81 | 37.73 ±0.04 | 37.78 ±0.00 | 19.95 ±0.05 | 30.52 ±0.08 | **43.89 ±0.01** |
| Defocus blur | 41.46 | 41.46 ±0.01 | 41.46 ±0.00 | 24.79 ±0.04 | 34.27 ±0.05 | **46.00 ±0.00** |
| Glass blur | 41.76 | 41.55 ±0.01 | 41.82 ±0.00 | 24.61 ±0.03 | 34.55 ±0.16 | **46.83 ±0.04** |
| Motion blur | 42.65 | 42.81 ±0.01 | 42.74 ±0.00 | 25.66 ±0.06 | 36.07 ±0.11 | **47.72 ±0.04** |
| Zoom blur | 34.46 | 34.25 ±0.00 | 34.44 ±0.00 | 20.74 ±0.05 | 28.44 ±0.07 | **39.07 ±0.02** |
| Snow | 40.13 | 40.47 ±0.00 | 40.23 ±0.01 | 25.02 ±0.06 | 33.98 ±0.03 | **46.30 ±0.05** |
| Frost | 40.70 | 41.83 ±0.08 | 40.80 ±0.00 | 23.43 ±0.05 | 34.09 ±0.01 | **46.78 ±0.01** |
| Fog | 42.50 | 42.67 ±0.00 | 42.47 ±0.00 | 24.95 ±0.05 | 36.37 ±0.04 | **47.61 ±0.06** |
| Brightness | 44.21 | 44.64 ±0.03 | 44.27 ±0.00 | 27.54 ±0.07 | 38.44 ±0.11 | **49.89 ±0.08** |
| Contrast | 40.44 | 40.14 ±0.03 | 40.44 ±0.00 | 23.89 ±0.04 | 33.68 ±0.05 | **45.22 ±0.00** |
| Elastic transform | 40.63 | 41.78 ±0.01 | 40.67 ±0.00 | 24.40 ±0.05 | 35.38 ±0.01 | **46.87 ±0.02** |
| Pixelate | 44.70 | 44.95 ±0.03 | 44.79 ±0.00 | 27.74 ±0.06 | 38.14 ±0.06 | **50.11 ±0.01** |
| JPEG compression | 43.05 | 42.87 ±0.04 | 43.05 ±0.00 | 26.04 ±0.05 | 36.29 ±0.10 | **48.27 ±0.02** |
| V21-C Average | 40.75 | 40.95 | 40.77 | 24.12 | 34.16 | **46.25** |

Table 22: mIoU comparison of MLMP and baseline methods on the P59 dataset, evaluated on both the original images and 15 corruption types.

| OVSS Backbone: NACLIP | | Adaptation Method | | | | |
|---|---|---|---|---|---|---|
| Dataset: P59 | No Adapt. | TENT | TPT | WATT | CLIPArTT | MLMP |
| Original | 28.23 | 28.73 ±0.02 | 28.26 ±0.01 | 16.55 ±0.05 | 24.60 ±0.00 | **31.95 ±0.02** |
| Gaussian noise | 21.53 | 21.49 ±0.01 | 21.59 ±0.01 | 11.27 ±0.04 | 16.42 ±0.03 | **24.84 ±0.00** |
| Shot noise | 22.35 | 22.31 ±0.01 | 22.26 ±0.00 | 11.80 ±0.04 | 17.47 ±0.00 | **25.62 ±0.01** |
| Impulse noise | 21.74 | 21.59 ±0.01 | 21.72 ±0.00 | 11.24 ±0.03 | 17.17 ±0.00 | **24.75 ±0.01** |
| Defocus blur | 25.42 | 25.14 ±0.00 | 25.41 ±0.00 | 14.31 ±0.04 | 20.71 ±0.00 | **27.95 ±0.03** |
| Glass blur | 25.03 | 24.70 ±0.01 | 25.01 ±0.00 | 13.93 ±0.03 | 20.63 ±0.00 | **27.66 ±0.05** |
| Motion blur | 26.11 | 26.00 ±0.02 | 26.13 ±0.01 | 15.13 ±0.03 | 21.72 ±0.00 | **29.12 ±0.03** |
| Zoom blur | 19.20 | 19.49 ±0.03 | 19.20 ±0.00 | 10.86 ±0.03 | 15.39 ±0.00 | **21.98 ±0.02** |
| Snow | 22.45 | 22.29 ±0.02 | 22.45 ±0.00 | 13.60 ±0.04 | 19.17 ±0.01 | **25.47 ±0.01** |
| Frost | 21.95 | 21.94 ±0.00 | 21.97 ±0.00 | 12.83 ±0.03 | 18.77 ±0.04 | **24.52 ±0.01** |
| Fog | 24.85 | 24.91 ±0.03 | 24.86 ±0.00 | 13.85 ±0.05 | 20.66 ±0.00 | **27.84 ±0.01** |
| Brightness | 27.39 | 27.80 ±0.01 | 27.42 ±0.00 | 15.55 ±0.04 | 23.56 ±0.00 | **30.63 ±0.01** |
| Contrast | 23.55 | 23.58 ±0.01 | 23.54 ±0.00 | 13.38 ±0.05 | 19.12 ±0.00 | **26.95 ±0.00** |
| Elastic transform | 23.30 | 23.86 ±0.02 | 23.30 ±0.01 | 12.88 ±0.04 | 20.28 ±0.00 | **27.16 ±0.01** |
| Pixelate | 27.61 | 27.55 ±0.01 | 27.62 ±0.00 | 15.84 ±0.05 | 23.47 ±0.00 | **31.23 ±0.01** |
| JPEG compression | 25.75 | 25.51 ±0.05 | 25.75 ±0.00 | 14.09 ±0.03 | 21.20 ±0.00 | **29.66 ±0.02** |
| P59-C Average | 23.88 | 23.88 | 23.88 | 13.37 | 19.72 | **27.03** |

Table 23: mIoU comparison of MLMP and baseline methods on the P60 dataset, evaluated on both the original images and 15 corruption types.

| OVSS Backbone: NACLIP | | Adaptation Method | | | | |
|---|---|---|---|---|---|---|
| Dataset: P60 | No Adapt. | TENT | TPT | WATT | CLIPArTT | MLMP |
| Original | 24.95 | 25.29 | 24.98 ±0.01 | 14.77 ±0.04 | 21.88 ±0.01 | **27.99 ±0.03** |
| Gaussian noise | 19.31 | 19.17 ±0.01 | 19.34 ±0.01 | 10.29 ±0.03 | 14.91 ±0.01 | **22.22 ±0.02** |
| Shot noise | 19.98 | 19.83 ±0.02 | 19.91 ±0.01 | 10.82 ±0.03 | 15.82 ±0.01 | **22.81 ±0.01** |
| Impulse noise | 19.56 | 19.27 ±0.01 | 19.54 ±0.00 | 10.34 ±0.03 | 15.58 ±0.00 | **22.26 ±0.01** |
| Defocus blur | 22.74 | 22.38 ±0.01 | 22.72 ±0.01 | 12.79 ±0.04 | 18.67 ±0.00 | **24.92 ±0.01** |
| Glass blur | 22.49 | 22.05 ±0.01 | 22.48 ±0.01 | 12.64 ±0.03 | 18.61 ±0.00 | **24.71 ±0.05** |
| Motion blur | 23.28 | 23.01 ±0.03 | 23.30 ±0.01 | 13.53 ±0.03 | 19.51 ±0.01 | **25.71 ±0.03** |
| Zoom blur | 17.42 | 17.58 ±0.03 | 17.42 ±0.00 | 9.95 ±0.03 | 14.07 ±0.01 | **19.70 ±0.02** |
| Snow | 20.06 | 19.80 ±0.01 | 20.06 ±0.01 | 12.29 ±0.04 | 17.20 ±0.02 | **22.64 ±0.02** |
| Frost | 19.71 | 19.61 ±0.01 | 19.73 ±0.01 | 11.59 ±0.03 | 17.02 ±0.04 | **22.02 ±0.01** |
| Fog | 22.20 | 22.09 ±0.02 | 22.20 ±0.01 | 12.50 ±0.03 | 18.53 ±0.01 | **24.89 ±0.00** |
| Brightness | 24.35 | 24.56 ±0.01 | 24.37 ±0.01 | 13.87 ±0.04 | 21.05 ±0.01 | **27.02 ±0.00** |
| Contrast | 21.09 | 21.00 ±0.03 | 21.08 ±0.01 | 12.04 ±0.03 | 17.24 ±0.01 | **24.17 ±0.02** |
| Elastic transform | 21.17 | 21.46 ±0.01 | 21.19 ±0.01 | 11.70 ±0.04 | 18.53 ±0.01 | **24.27 ±0.02** |
| Pixelate | 24.52 | 24.33 ±0.01 | 24.54 ±0.01 | 14.17 ±0.04 | 21.01 ±0.01 | **27.48 ±0.00** |
| JPEG compression | 22.99 | 22.65 ±0.02 | 24.54 ±0.01 | 12.65 ±0.03 | 19.05 ±0.01 | **26.24 ±0.01** |
| P60-C Average | 21.39 | 21.25 | 21.49 | 12.08 | 17.79 | **24.07** |

Table 24: mIoU comparison of MLMP and baseline methods on the CityScapes dataset, evaluated on both the original images and 15 corruption types.

| OVSS Backbone: NACLIP | Adaptation Method | | | | |
|---|---|---|---|---|---|
| Dataset: CityScapes | No Adapt. | TENT | TPT | WATT | MLMP |
| Original | 29.49 | 30.54 ±0.04 | 29.57 ±0.01 | 20.77 ±0.02 | **33.35** ±**0.03** |
| Gaussian noise | **16.14** | 15.00 ±0.02 | 15.94 ±0.01 | 10.19 ±0.01 | 15.32 ±0.00 |
| Shot noise | 20.18 | 19.38 ±0.01 | 20.15 ±0.01 | 12.41 ±0.02 | **20.57** ±**0.04** |
| Impulse noise | **16.37** | 14.59 ±0.02 | 16.35 ±0.01 | 8.56 ±0.02 | 15.76 ±0.05 |
| Defocus blur | 23.34 | 23.50 ±0.02 | 23.46 ±0.01 | 15.63 ±0.02 | **24.86** ±**0.08** |
| Glass blur | 24.41 | 24.04 ±0.04 | 24.29 ±0.00 | 15.44 ±0.01 | **25.70** ±**0.02** |
| Motion blur | 24.73 | 24.98 ±0.00 | 24.91 ±0.01 | 14.21 ±0.03 | **26.02** ±**0.01** |
| Zoom blur | 14.08 | **14.57** ±**0.06** | 14.08 ±0.01 | 5.85 ±0.02 | 14.04 ±0.04 |
| Snow | 19.88 | 20.14 ±0.01 | 19.90 ±0.01 | 9.27 ±0.03 | **22.23** ±**0.03** |
| Frost | 16.78 | 16.47 ±0.02 | 16.78 ±0.01 | 10.69 ±0.01 | **17.12** ±**0.07** |
| Fog | 22.47 | 22.87 ±0.05 | 22.25 ±0.01 | 14.24 ±0.01 | **23.98** ±**0.03** |
| Brightness | 28.45 | 29.30 ±0.01 | 28.47 ±0.01 | 19.99 ±0.02 | **31.88** ±**0.04** |
| Contrast | 16.10 | 16.35 ±0.03 | 16.07 ±0.01 | 11.05 ±0.01 | **17.04** ±**0.00** |
| Elastic transform | 29.14 | 30.16 ±0.02 | 29.02 ±0.01 | 19.99 ±0.03 | **32.86** ±**0.00** |
| Pixelate | 28.67 | 29.59 ±0.01 | 28.69 ±0.01 | 18.30 ±0.01 | **32.72** ±**0.04** |
| JPEG compression | 23.69 | 23.62 ±0.01 | 23.67 ±0.01 | 15.98 ±0.02 | **25.26** ±**0.03** |
| CityScapes-C Average | 21.63 | 21.64 | 21.60 | 13.45 | **23.02** |

Table 25: Detailed ACDC mIoU (reference vs. adverse views). All experiments use NACLIP as the OVSS model. MLMP consistently improves over TENT and the non-adapted baseline across all weather conditions.

| OVSS: NACLIP | Adaptation Method | | |
|---|---|---|---|
| Condition | No Adapt. | TENT | MLMP |
| Fog (ref) | 24.80 | 26.52 ±0.03 | **31.28** ±**0.03** |
| Fog | 23.88 | 26.89 ±0.04 | **33.33** ±**0.04** |
| Night (ref) | 24.95 | 26.54 ±0.04 | **28.81** ±**0.10** |
| Night | 22.12 | 24.17 ±0.00 | **24.76** ±**0.03** |
| Rain (ref) | 24.79 | 26.62 ±0.00 | **31.10** ±**0.03** |
| Rain | 23.86 | 26.84 ±0.04 | **32.44** ±**0.04** |
| Snow (ref) | 22.10 | 24.27 ±0.04 | **28.22** ±**0.02** |
| Snow | 23.54 | 27.25 ±0.05 | **30.59** ±**0.03** |
| Average (all ref) | 24.16 | 25.99 | **29.85** |
| Average (all domains) | 23.35 | 26.29 | **30.28** |

