# OpenReview forum: "Test-Time Adaptation of Vision-Language Models for Open-Vocabulary Semantic Segmentation"
_NeurIPS.cc/2025/Conference — NeurIPS 2025 poster_

### Official Review · Reviewer_Fuid · 2025-06-13

**Clarity:** 2
**Significance:** 2
**Originality:** 3
**Rating:** 5
**Confidence:** 3

**Summary:**

The paper introduces a novel method for test-time adaptation of contrastive VLMs for open-vocabulary semantic segmentation. The method can be summarized in two steps:

1. Update/adaptation: the authors tune the layer norm modules by minimizing the entropy of the distribution resulting from a (softmaxed) dot-product between prompt embeddings and image features. For this, they use (a) features from the last 75% of ViT blocks, averaged together, and (b) 7 prompt templates. They then calculate loss terms for each template and average them.
2. Inference: the authors predict the segmentation map using the same formulation as CLIP zero-shot for image classification but on a spatial token basis. They use the same seven prompts and features as in the adaptation step, but they re-weight the visual features based on their batch-wise entropy.

The authors demonstrate that their method significantly improves performance compared to the non-adapted baseline on seven datasets, both in its "clean" version and when corrupted using the same image corruptions as those in ImageNet-C.

Their method also performs significantly better than other test-time adaptation strategies originally conceived for image classification but adapted to open-vocabulary semantic segmentation.

**Questions:**

1. The adaptation/inference loop is not fully clear to me. My understanding is that: you pick two test samples, perform 10 steps of adaptation, predict the masks of both samples, and then restore the model. Is this correct? If yes, why was a batch size of two chosen? Intuitively, I would expect that adapting for a single sample would work better.
	1. Would a larger (or smaller) batch size lead to better results?
	2. It could be interesting to explore an online version of the method, i.e., rather than discarding the updates, use them as a starting point for the next batch or sample. Would the performance degrade or improve significantly, as in [1]?
2. WATT and CLIPArTT perform significantly worse than a non-adapted baseline. Do you have any insights on why this happens?
3. Since the multi-prompt loss alone improves performance by a good margin, it could be interesting to see if further improvements can be achieved with longer prompts using either Long-CLIP [2] or TULIP [3].

---

- [1] Test-Time Training with Self-Supervision for Generalization under Distribution Shifts, Yu Sun et al., ICML 2020.
- [2] Long-CLIP: Unlocking the Long-Text Capability of CLIP, Beichen Zhang et al., ECCV 2024
- [3] TULIP: Token-length Upgraded Clip, Ivona Najdenkoska, Mohammad Mahdi Derakhshani et al., ICLR 2025

**Ethical Concerns:**

["NO or VERY MINOR ethics concerns only"]

**Final Justification:**

The authors addressed all my concerns and provided new and insightful experiments. I am thus increasing my score to 5.

**Limitations:**

The authors appropriately discussed the method's limitations.

**Paper Formatting Concerns:**

Not applicable.

**Quality:**

3

**Strengths And Weaknesses:**

### Strengths

1. The paper is reasonably well-written and easy to follow. The method is novel and tackles a previously understudied area.
2. The method significantly outperforms non-adapted backbones and existing baselines adapted for open-vocabulary semantic segmentation. These improvements are validated across various datasets, corruptions, model sizes, and model families, the latter two explored in the supplementary materials.
3. The methodological choices are explained through extensive ablations.

---

### Weaknesses

1. Some parts of the paper could be further polished and are a bit unclear.
	1. In Figure 1, I would recommend adding an x-axis label for the figure on the left and specifying in the caption what backbone is being used and what V20 means (it's not immediately clear that this refers to Pascal VOC). Furthermore, it's not clear how the variance in the left plot was computed. Is it computed over a single example? Or multiple?
2. The paper claims that using multiple prompts prevents degenerate collapse and ensures that the model yields segmentations that are consistent across diverse linguistic formulations but never shows a case of degenerate collapse with a single prompt. I am also skeptical of the second claim, as to my understanding, the authors always use the same set of prompts for both adaptation and inference across the paper's experiments.
3. In Figure 4, it could also be interesting to see the performance with no adaptation to see how much a single prompt improves over the baseline.
4. On line 212, you mention the GTAV dataset but do not cite it.
5. On line 155, the paper states that the hypothesis that layers contain complementary information is validated in Figure 1b. However, this figure only compares the overall performance of MLMP to a non-adapted baseline and does not provide insights into this specific claim.

---

> ### Author Rebuttal · Authors · 2025-07-31
>
> We thank the reviewer for their thoughtful and constructive feedback. We are encouraged that they found our method novel, well-motivated, and clearly presented, and that they appreciated the strong empirical results across diverse datasets, corruption types, and model families. We also thank the reviewer for highlighting the value of our extensive ablations and recognizing that our work addresses an understudied area. Below, we respond to each of the reviewer’s comments in detail.
>
> **A1. Improving Figure 1**
>
> The variance is computed over all samples in the V20-C benchmark, which includes all corruption types. We appreciate the reviewer’s feedback and will incorporate all these clarifications to improve the clarity and completeness of the figure.
>
> **A2. Clarifications on Multi-Prompt Robustness Claims**
>
> To illustrate how multiple prompts help prevent degenerate collapse, we conducted an experiment where we ran adaptation using either all seven templates or only a single one and averaged them, using the same 10 images for efficiency. The results are summarized in the table below.
> Template|No Adapt.|50 Steps|100 Steps|200 Steps|
> |-|-|-|-|-|
> |All Templates|38.20|65.42|64.75|65.74|
> |Average of single template|37.63|59.79|58.60|56.88|
>
> On V20-C, single-template runs often show unstable behavior or steep drops as we can see thanks to the average, in contrast, the multi-template setup maintains high performance, with only a slight decline. These results support our claim that using multiple prompts acts as a regularizer and mitigates collapse during adaptation.
>
> **A3. Improving Figure 4**
>
> We will update this figure in the final version by adding the performance with no adaptation (the quantitative results are already in the main table).
>
> **A4. Missing Citation for GTAV**
>
> We will add the appropriate citation for the GTAV dataset in the final version.
>
> **A5. Typo in Line 155**
>
> Thank you for pointing this out. The reference to Fig. 1b in that sentence is a typo. The intended evidence for the complementary nature of intermediate layers is in Table 1, which reports the ablation on fusion strategies, and in Figure 3, which visualizes the learned layer-wise confidence weights. We will correct the sentence and update the citation accordingly in the final version.
>
> **A6. Clarification on Adaptation Loop and the Effect of Batch Size**
>
> The reviewer’s understanding of the adaptation/inference loop is correct: we adapt the model using a batch of two test samples for 10 steps, predict the masks for both samples, and then restore the original model parameters.
>
> We chose a batch size of 2 mainly for computational efficiency; it provides roughly 2× faster adaptation compared to using a batch size of 1, which was important given the need to evaluate multiple adaptation methods across several datasets.
>
> While larger batch sizes can be beneficial in classification due to entropy bias correction, this is less relevant in segmentation, where supervision occurs at the pixel/token level. In fact, increasing the batch size may require additional tuning of hyperparameters and can hurt performance due to over-smoothing or weaker adaptation signals.
>
> To address your intuition that a batch size of 1 may work better, we confirm this in Appendix Table 3. As shown below, our method performs well with a batch size of 1, supporting the robustness of the approach across batch configurations:
> |Dataset|No Adapt.|TENT (bs=1)|TENT (bs=2)|MLMP (bs=1)|MLMP (bs=2)|
> |-|-|-|-|-|-|
> |V20|75.91|76.20 ± 0.00|77.00 ± 0.04|84.68 ± 0.00|83.76 ± 0.00|
> |V20-C|69.01|68.50|69.33|78.41|77.58|
>
> We will update the paper to make this clearer and explicitly reference the ablation results to improve transparency.
>
> **A7. Episodic vs Online Setting**
>
> We thank the reviewer for the insightful suggestion regarding an online adaptation setting. We experimented with this setting using the exact same hyperparameters as our main method (i.e., learning rate and adaptation steps), but found that both our method and TENT eventually collapsed: while performance initially improved, it degraded after a few batches and dropped significantly thereafter.
>
> Motivated by the reviewer’s reference to [1], we further explored the online setting by reducing the number of adaptation steps and lowering the learning rate. The results are as follows:
> |Dataset|No Adapt.|Episodic: TENT|Episodic: MLMP|Online: TENT|Online: MLMP|
> |-|-|-|-|-|-|
> |V20|75.91 ± 0.00|77.00 ± 0.04|83.76 ± 0.00|76.44 ± 0.21|79.19 ± 0.45|
> |V20-C|69.01|69.33|77.58|69.32|71.77|
>
> We observe that while the online variant improves over baselines, it still underperforms compared to the reset-based (episodic) setting. We believe this is primarily due to error accumulation and model drift, as documented in prior work showing that some recent classification TTA methods exhibit performance degradation in an online setting—even when the distribution is static [2]. This issue is particularly pronounced in semantic segmentation, where dense, spatial predictions make the model more sensitive to incorrect updates, and entropy minimization can amplify early misclassifications. Over time, this leads the model to drift away from its well-initialized source parameters, reducing its ability to generalize across the target domain.
>
> Episodic (reset) adaptation avoids the runaway effects of carrying over errors, which is why it tends to be safer [2]. In contrast, fully online adaptation—though attractive for its potential to continuously refine the model—must confront these challenges.
>
> We believe that online adaptation in segmentation can be improved through several future directions: (1) introducing occasional resets or using exponential moving average (EMA) of model weights to limit drift and reduce accumulated errors; (2) filtering high-entropy samples or gradients, which is particularly important in segmentation due to its spatial granularity and sensitivity to local noise; and (3) incorporating an auxiliary self-supervised objective, such as rotation prediction [1], to provide a more reliable adaptation signal—though this may come with increased computational cost. Notably, the fact that our method works effectively with very few test samples and does not rely on accumulating state makes it efficient for real-world applications.
>
> We will include a discussion of these findings and potential improvements in the final version.
>
> [1] Test-Time Training with Self-Supervision for Generalization under Distribution Shifts, ICML 2020.
>
> [2] On Pitfalls of Test-time Adaptation, ICLR 2023
>
> **A8. Insights on the Underperformance of WATT and CLIPArTT**
>
> Thank you for the observation. Indeed, as shown in the table below, our own variant of MLMP using a pseudo-labeling loss performs worse than the non-adapted baseline, consistent with the behavior observed for WATT and CLIPArTT. We also show in Table 3 of the paper that parameter averaging, as done in WATT, does not improve performance in our setup either.
>
> We believe this highlights a key difference between segmentation and classification tasks. In segmentation, pseudo-labeling relies on confidently predicting a hard label for every pixel, which can amplify noise and lead to unstable adaptation, especially under distribution shift. In contrast, entropy minimization provides a smoother learning signal by encouraging confident predictions without committing to hard labels, making it more effective in this context.
> |Dataset|No Adapt.|MLMP|MLMP PseudoLabel|
> |-|-|-|-|
> |V20|75.91|83.76 ± 0.00|70.06 ± 0.18|
> |V21|45.12|50.78 ± 0.02|43.78 ± 0.11|
> |V20-C|69.01|77.58|56.31|
> |V21-C|40.75|46.25|35.74|
>
> We will clarify these insights in the paper to better explain why some adaptation strategies from classification may not transfer well to segmentation.
>
> **A9. Experience with Longer Prompts**
>
> Thank you for the suggestion. We agree that exploring longer prompts, such as those enabled by models like Long-CLIP or TULIP, can be an interesting direction. To investigate this, we generated extended templates based on our 7 class-agnostic prompts using ChatGPT, as well as extended class names, where each class was described in natural language. For example, "a photo of the large {class}." becomes "A photograph of a very large {}, where the size of the object dominates the frame or is shown in contrast to smaller elements to emphasize its scale." and “aeroplane” becomes “aeroplane: a powered flying vehicle with fixed wings and engines, designed to transport people or cargo through the air over long distances”
>
> As shown in the table below, MLMP consistently improves performance over the Long-CLIP baseline across all variants. However, using extended templates or extended class descriptions did not lead to further improvements over the base prompts. This suggests that while MLMP is robust and effective even with longer inputs, simply extending prompt length does not guarantee performance gains, possibly due to the complexity or ambiguity introduced by verbose descriptions.
> |Dataset|LongCLIP|LongCLIP + MLMP|Extended Templates|Extended Templates + MLMP|Extended Classes|Extended Classes + MLMP|
> |-|-|-|-|-|-|-|
> |V20|39.64|54.45 ± 0.09|35.55|53.43 ± 0.06|25.66|43.92 ± 0.04|
> |V21|16.83|19.54 ± 0.02|15.87|19.15 ± 0.04|11.27|15.25 ± 0.03|
> |V20-C|29.36|48.71|26.92|48.90|21.09|39.01|
> |V21-C|13.64|18.57|12.98|18.27|10.23|14.66|
>
> We will add this analysis and discussion to the paper to provide a clearer picture of MLMP’s behavior under extended prompts.

---

> > ### Comment · Reviewer_Fuid · 2025-08-03
> >
> > Thank you for your hard work and the detailed rebuttal! The reported results are very interesting.
> >
> > I am particularly surprised that an online setting does not improve performance. I assume that the reported results are for the original (clean) Pascal VOC dataset, correct? If so, did the authors also try this with a single corruption type from VOC-C, e.g., snow/frost/fog? I would expect it to potentially provide an advantage in such a setup, as it better aligns with an online setting.
> >
> > As for the longer prompts, it is surprising to me that the extended templates perform significantly worse compared to the baseline. However, it is interesting to note that the absolute improvement of MLMP over the baseline is larger in this setup, suggesting that a “prompt search” might be worth exploring in the future.

---

> > > ### Author Response · Authors · 2025-08-04
> > >
> > > Thank you for your thoughtful comments and for engaging with our work in such depth.
> > >
> > >
> > > Regarding the online setting, the results in the rebuttal are reported on Pascal VOC 20 (clean), referred to as V20, and on V20-C, which is the average over all corruption types applied to Pascal VOC 20. To address your question more specifically, we include below a breakdown on selected corruptions from V20-C:
> > >
> > > |Dataset|No Adapt.|Episodic: TENT|Episodic: MLMP|Online: TENT |Online: MLMP|
> > > |-|-|-|-|-|-|
> > > |V20 (Original)|75.91|77.00 ± 0.04|83.76 ± 0.00|76.44 ± 0.21|79.19 ± 0.45|
> > > |Snow|71.49|71.04 ± 0.05|79.53 ± 0.05|71.91 ± 0.14|73.21 ± 0.20|
> > > |Frost|65.38|67.01 ± 0.02|73.20 ± 0.07|65.58 ± 0.06|66.83 ± 0.19|
> > > |Fog|70.69|70.54 ± 0.07|79.81 ± 0.06|70.92 ± 0.10|74.09 ± 0.42|
> > > |V20-C Average|69.01|69.33|77.58|69.32|71.77|
> > >
> > >
> > >
> > > As shown in the table, online adaptation of MLMP improves upon TENT in both clean and corrupted settings. However, the episodic variant of MLMP consistently outperforms its online counterpart. As we noted in the rebuttal, we believe that the episodic (reset) adaptation helps avoid the compounding of errors during adaptation, which is particularly important in dense prediction tasks such as segmentation.
> > >
> > > ---
> > >
> > > Regarding the use of longer prompts, we agree with your assessment. In the LongCLIP paper, similar prompt templates are used for image classification, while extended prompts are mainly explored in image generation and retrieval tasks. We believe that using longer prompts effectively for segmentation requires more targeted design (e.g., including visual attributes like shape, color, or spatial cues). As you suggested, exploring this direction via prompt search or learning could be a valuable avenue for future work.

---

> > > > ### Comment · Reviewer_Fuid · 2025-08-06
> > > >
> > > > Thank you for your response and providing a per-corruption breakdown. The results are indeed very convincing, and I believe they support your hypothesis that errors are likely to compound in dense tasks such as segmentation.
> > > >
> > > > I will either maintain or improve my rating upon further consideration and taking into account feedback from the other reviewers. Thank you again for engaging in the discussion!

---

> > > > > ### Author Response · Authors · 2025-08-06
> > > > >
> > > > > Thank you for your follow-up and for taking the time to engage so thoughtfully with our work. We appreciate your careful consideration and constructive suggestions throughout the process. Your feedback has helped us strengthen the clarity and scope of the work. We would be happy to further elaborate on any remaining questions or suggestions you might have.

---

### Official Review · Reviewer_tiCb · 2025-06-30

**Clarity:** 3
**Significance:** 2
**Originality:** 2
**Rating:** 4
**Confidence:** 4

**Summary:**

This paper proposes a novel test-time adaptation (TTA) method, MLMP, for adapting vision-language models (VLMs) to segmentation tasks during inference. MLMP performs multi-level and multi-prompt (MLMP) entropy minimization by integrating features from intermediate layers of the vision encoder and optimizing both global CLS tokens and local pixel-level representations using different textual prompt templates. MLMP is a plug-and-play module that can be applied to any segmentation network without requiring additional training data or labels, and it remains effective even when only a single test sample is available. A comprehensive benchmark suite for open-vocabulary semantic segmentation (OVSS) TTA is constructed, covering seven segmentation datasets, 15 common corruptions, and 82 diverse testing scenarios, providing a standardized and thorough platform for future OVSS TTA research.

**Questions:**

Please answer the questions in Weakness and refute the weaknesses I pointed out about Feature Fusion and Prompt Templates.

**Ethical Concerns:**

["NO or VERY MINOR ethics concerns only"]

**Final Justification:**

Thank you for the author's response. I maintain my original rating.

**Limitations:**

The authors discussed their limitations of their work.

**Quality:**

3

**Strengths And Weaknesses:**

**Strengths**:
1、The focus on test-time adaptation for VLMs is of high practical relevance.
2、Theoretical analysis demonstrates that using multiple prompt templates helps reduce the variance of the loss gradient, laying a foundation for future research.
3、The experiments are extensive and thorough, supporting the proposed method from multiple perspectives.
4、The proposed method achieves significant performance improvements across multiple datasets.

**Weaknesses**:

1、Feature fusion across layers and the use of image-level entropy as a loss are relatively common techniques, making this part of the method less innovative.
2、In the Experimental Setup section, it is stated that the adaptation process is conducted over 10 iterations with resets. However, as far as I know, TTA is typically evaluated under a one-epoch protocol to test effectiveness. How does the proposed method perform under that setting?
3、How exactly does the method support open-vocabulary settings? Line 120 mentions that a set of concepts C_k in C must be provided during inference. Does this imply that the model can only segment categories within the predefined set C? It is suggested to provide more detailed clarification to highlight the method's capability in handling open-vocabulary segmentation.
4、The method appears sensitive to the choice of prompt templates. Although multi-prompt optimization helps mitigate this issue to some extent, selecting more effective prompt templates to further improve performance remains an open challenge that warrants deeper investigation.

---

> ### Author Rebuttal · Authors · 2025-07-31
>
> We thank the reviewer for their thoughtful and constructive feedback. We are pleased that they acknowledged the practical significance of adapting vision-language models during inference, appreciated our theoretical insights on prompt-based variance reduction, and recognized the strength of our experimental results. In the following, we address the reviewer’s comments point by point.
>
>
> **A1. Novelty of Feature Fusion and Image-Level Loss**
>
> Thank you for the comment. While feature fusion across layers is indeed common in standard segmentation models (e.g., U-Net, HRNet, FPN), its use in TTA, especially in the context of OVSS, has not been explored to our knowledge. In our method, we go beyond basic fusion by introducing adaptive layer weighting, which allows the model to selectively emphasize features based on their relevance during adaptation. This is a key contribution that distinguishes our approach.
>
> As for the image-level entropy loss, we note that this is not a common technique in segmentation, and especially rare in OVSS. Unlike pixel-level entropy minimization used in methods like TENT, we apply entropy regularization at the image-level representation (CLS token), encouraging more confident global predictions in a zero-shot setting. To our knowledge, this is the first use of such a loss in OVSS TTA.
>
> We believe the overall design brings a novel perspective to this recent problem, a view that is also echoed in the feedback from R-Fuid.
>
> **A2. Evaluation Under One-Iterate TTA Protocol**
>
> The number of adaptation iterations in TTA varies across the literature. While some works evaluate under a one-iteration (step) protocol, many others use multiple iterations. For example, the original TTT paper (Sun et al., 2020) as well as more recent vision-language TTA methods like CLIPArTT and WATT commonly use 10 adaptation iterations. Our choice to follow this convention is aligned with these prior works. For fairness, we use the same hyperparameters across all experiments and baseline methods.
>
> Nonetheless, we agree it is valuable to assess performance under a single-iteration protocol. As shown in the table below, MLMP still improves over the baseline and outperforms TENT even with just one iteration, which demonstrates the effectiveness of our method even under a stricter setting.
> |Dataset|No Adapt.|TENT|**MLMP**|
> |-|-|-|-|
> |V20|75.91|76.80±0.01|**79.49±0.01**|
> |V21|45.12|45.36±0.00|**49.97±0.00**|
> |V20-C|69.01|70.15|**73.38**|
> |V21-C|40.75|41.02|**45.54**|
>
> We will clarify this setup in the main paper and include the single-iteration results to make the evaluation more comprehensive.
>
> **A3. Open-Vocabulary Capability**
>
> We thank the reviewer for this insightful question. The set of concepts C_k ⊂ C mentioned in Line 120 is provided at inference time and can be arbitrary—there is no restriction to a fixed or predefined vocabulary. This flexibility is what enables open-vocabulary segmentation: the model is not trained on specific categories and can adapt to any set of text labels at test time.
>
> In our experiments, we follow the evaluation protocol used in prior OVSS works (e.g., NACLIP [1] ), which adopt a predefined set C purely for benchmarking purposes. However, this set can be changed at any time, and the model is capable of segmenting new, unseen categories that were not present during training.
>
> We will clarify this aspect in the paper to better emphasize that our method fully supports open-vocabulary segmentation by design.
>
> [1] Pay Attention to Your Neighbours: Training-Free Open-Vocabulary Semantic Segmentation, WACV 2025
>
> **A4. Selection of Prompts**
>
> Thank you for the observation. We agree that prompt selection is an important aspect in vision-language models. As noted in Appendix Section C, we did not tune prompt templates for performance; instead, we used the general-purpose templates proposed by the original CLIP authors. These prompts are not tailored to specific image content, which helps ensure fairness and broad applicability.
>
> While our multi-prompt strategy helps mitigate sensitivity to any single template, we agree that exploring more effective or adaptive prompt selection strategies could further improve performance. However, such approaches often come with additional computational and memory costs, and we consider them complementary to our method. We view this as a promising direction for future work.

---

> > ### Comment · Reviewer_tiCb · 2025-08-05
> >
> > Thank you for the author's response. The A1 provided by the author did not address my concerns, so I will maintain my original rating.

---

> > > ### Author Response · Authors · 2025-08-05
> > >
> > > We sincerely thank the reviewer for their time and engagement. To briefly recap, our paper introduces (1) an uncertainty-aware fusion strategy that dynamically adjusts per-layer contributions at test time, and (2) a multi-prompt integration scheme within the adaptation loss: both tailored for the novel and largely unexplored setting of test-time adaptation for open-vocabulary semantic segmentation. We also contribute a comprehensive benchmark for this task, incorporating both synthetic and realistic distribution shifts.
> > > Regarding the comment on feature fusion and entropy loss: while these components may have appeared in other contexts, to the best of our knowledge, they have not been tailored or applied in the OVSS-TTA setting. We believe their integration here is non-trivial and well-justified by our empirical results.
> > > We would be happy to clarify any remaining points. We also note that your final comment seemed focused on the novelty of two individual components (layer fusion and image-level entropy loss), which are not the core contributions, rather than the overall contribution of the method and benchmark which, we hope, is not in question. Please don’t hesitate to let us know if any aspect remains unclear or would benefit from further elaboration.

---

### Official Review · Reviewer_rgyA · 2025-07-02

**Clarity:** 3
**Significance:** 3
**Originality:** 3
**Rating:** 4
**Confidence:** 4

**Summary:**

The paper introduces MLMP (Multi-Level Multi-Prompt), a test-time adaptation (TTA) framework for open-vocabulary semantic segmentation (OVSS). MLMP focuses on dense prediction tasks by:
- Integrating features from intermediate vision-encoder layers (e.g., ViT layers) to capture complementary, shift-resilient cues, with uncertainty-aware weighting based on entropy.
- Minimizing entropy across multiple text-prompt templates at both global and local levels to leverage prompt sensitivity as a robust adaptation signal.
The authors also establish an OVSS TTA benchmark to demonstrate its effectiveness in adapting to domain shifts during inference.

**Questions:**

see weakness

**Ethical Concerns:**

["NO or VERY MINOR ethics concerns only"]

**Final Justification:**

The author's response solves my concern, and the supplemented table makes the paper more readable. Thus I raise my score to 4: Borderline accept.

**Limitations:**

yes

**Quality:**

3

**Strengths And Weaknesses:**

Strength:
- The writing and architecture figure is clear and easy to follow.
- The observation revealed in Fig 1.(a) is interesting. And the ablation accordingly (Fig 3) is vivid.

Weakness:
- Weak Motivation:
I question the necessity of the OVSS TTA task. Most domain shifts in the authors' benchmark (e.g., blur, noise) are artificially created and lack practical significance. Such image shifts can be easily resolved through training-time data augmentation or post-processing. Modern segmentation frameworks no longer emphasize these augmentations because these shifts are already considered solved problems. Thus, re-investigating adaptation for them is of limited value. The authors should identify more compelling TTA tasks or scenarios.

- Missing Comparison:
Beyond zero-shot (ovss are considered as zss), the authors should compare with classical few-shot segmentation (fss) methods. These solutions also address domain shifts by adapting with a few samples without requiring training. The lack of comparison and discussion with such methods undermines the justification for OVSS TTA's necessity and validity.


- Overclaim of generalization ability:
The claim that MLMP applies to "any OVSS framework" is incorrect. It clearly only works for pixel-to-text similarity matching frameworks, not query-based ones (e.g., SAM, SEEM) that use decoders for segmentation. Additionally, validation is limited to NACLIP, no other CLIP-based or non-CLIP frameworks are tested or mentioned. The "any OVSS framework" claim is obviously an overstatement.

---

> ### Author Rebuttal · Authors · 2025-07-31
>
> We thank the reviewer for their time and feedback. We are glad that they found the writing clear and appreciated the observations motivating our MLMP design. In the following, we address each of the reviewer’s comments in detail.
>
> **A1. General Motivation behind TTA for OVSS**
>
> We appreciate the reviewer’s concern and would like to clarify the motivation behind the OVSS-TTA task. The main goal is to evaluate whether test-time adaptation can offer consistent improvements across a broad spectrum of potential shifts. As also noted by Reviewer t4bA, we believe the task is well motivated, and we agree with Reviewer tiCb on its high practical relevance, particularly given the unique challenges of zero-shot OVSS, where models are especially sensitive to distribution shifts due to the lack of supervised adaptation. This sensitivity is evident in Table 4 of the main paper, where the non-adapted model's performance drops from 75.91 mIoU on V20 (original) to 69.01 mIoU on V20-C (common corruptions), underscoring that even robust foundation models remain vulnerable to distribution shifts in the absence of test-time adaptation.
>
> While some of our benchmark shifts are synthetic (e.g., blur, noise), this follows standard practice for robustness evaluation, such as in ImageNet-C, and provides a controlled, reproducible setting to test the effectiveness of adaptation strategies. This is especially valuable in zero-shot scenarios, where the test distribution is inherently unknown.
>
>
> To demonstrate that our approach goes beyond synthetic corruptions, we include results on more realistic domain shifts. In our response to t4bA-A3, we present evaluations on ACDC, which features real-world weather-related conditions, and GTA5, a distinct synthetic domain with different scene statistics. We also highlight intra-dataset variations within Cityscapes (e.g., lighting, camera viewpoints, and location diversity). Furthermore, our method improves performance not only under distribution shift but also on the original (unshifted) datasets (Table 4), which underscores the general utility of test-time adaptation in OVSS.
>
> In real-world settings, it is unlikely that a model can fully account for all possible unseen target distributions. This is due to several factors: (1) the space of potential corruptions and environmental shifts is vast and essentially unbounded; (2) training on such a wide range of conditions would require extensive labeled and diverse data, which is often costly or impractical to obtain; and (3) the distribution encountered at deployment is typically unknown and may evolve over time, especially in open-world scenarios like OVSS.
>
> These constraints make TTA not only relevant but necessary, as it allows models to adjust to novel domains without requiring retraining or labeled data.
>
> **A2. Comparison with Few-Shot**
>
> Thank you for raising this point. While FSS methods do allow for adaptation using a small number of labeled examples, we believe they are fundamentally different from the TTA setting we explore in OVSS.
>
> First, most FSS methods assume access to a clean support set with annotated masks at inference time. In contrast, OVSS operates in a fully zero-shot setting, without any class-specific labels or examples at test time. The set of target categories is also unknown and potentially large, making it impossible to construct or assume a meaningful support-query structure.
>
> Moreover, while FSS methods do not require training-time fine-tuning, they still rely on labeled masks at inference, which can be difficult to obtain for open-vocabulary segmentation tasks, especially when dealing with arbitrary or long-tail classes.
>
> For these reasons, we believe that FSS and TTA operate under different assumptions and are not directly comparable. However, we appreciate the suggestion and will include a discussion of this distinction in the revised version of the paper.
>
> **A3. Clarification on the Generality of MLMP Across OVSS Frameworks**
>
> We would like to clarify that we do evaluate MLMP across multiple similarity-based OVSS frameworks to assess its generalization ability, as mentioned in lines 282–283 of the paper. Specifically, we test MLMP with (1) different OVSS methods (Table 6 in Appendix), and (2) different vision-language models, namely SigLIP-2 (Table 7 in Appendix) or LongCLIP (please see R-Fuid-A9) and (3) different backbones (Tables 4 and 5 in Appendix). These experiments demonstrate that MLMP is robust across architectures, feature extractors, and vision-language model variants within this class of methods.
>
> As stated in line 117, our method is specifically tailored for the OVSS framework based on contrastive VLMs, which covers most existing approaches in open-vocabulary semantic segmentation. We will ensure this point is stated more clearly in the revised manuscript.

---

> > ### Comment · Reviewer_rgyA · 2025-08-02
> > **Still have concerns about FSS**
> >
> > Thank you for the authors’ response!
> > However, the concerns regarding FSS methods remain unresolved.
> >
> > In the rebuttal, the authors mainly clarify the distinction between FSS and OVSS, but that is not my concern. What I care about is the difference between the authors’ proposed OVSS-TTA and FSS. FSS requires a clean support set with annotated masks at inference time, while OVSS-TTA also needs the same data for its test-time training. Therefore, it is entirely valid to treat the test-time data used by OVSS-TTA as the support set and directly compare it to FSS.

---

> > > ### Author Response · Authors · 2025-08-04
> > >
> > > **Comparing Our Setting with Few-Shot**
> > >
> > > We appreciate the reviewer’s follow-up and the opportunity to further clarify our methodological setting. In response to the statement, *“FSS requires a clean support set with annotated masks at inference time, while OVSS-TTA also needs the same data for its test-time training,”* we would like to respectfully emphasize that our method operates under the paradigm of **fully unsupervised test-time adaptation (TTA)** [1], which is **fundamentally different** from the supervised nature of **few-shot segmentation (FSS)**.
> > >
> > > More specifically, our OVSS-TTA setting, as well as the broader TTA paradigm, **does not access any labeled data, ground-truth masks or any other supervision**. In contrast, few-shot segmentation methods explicitly assume access to **a small set of labeled examples at inference time (support set)**, and adapt using supervised loss or prototype-based matching. Therefore, our method addresses a much more challenging setting, requiring adaptation from **unlabeled test-time input only**. Our setting also differs from **test-time training (TTT)**, which requires access to source data and training-time architectural or objective modifications (e.g., auxiliary self-supervised tasks).
> > >
> > > To further clarify these distinctions, we provide the following table comparing the key characteristics of different learning paradigms, including access to source data, the need for labeled supervision, whether an additional training phase is required, and the types of losses used:
> > >
> > >
> > > | **Setting**            | **Source data**       | **Target data**            | **Train loss**                                        | **Test loss**                 |
> > > |------------------------|------------------------|-----------------------------|--------------------------------------------------------|-------------------------------|
> > > | **Zero-Shot Inference**| ❌                     | $x^t$                       | ❌                                                     | ❌                            |
> > > | **Few-Shot Learning**  | ❌                     | $x^t, y^t$ (few)            | $\mathcal{L}(x^t, y^t)$ or prototype-based             | ❌                            |
> > > | **Test-Time Training** | $x^s, y^s$             | $x^t$                       | $\mathcal{L}(x^s, y^s) + \mathcal{L}_{aux}(x^s)$       | $\mathcal{L}_{aux}(x^t)$     |
> > > | **Fully TTA (Ours)**   | ❌                     | $x^t$                       | ❌                                                     | $\mathcal{L}_{unsup}(x^t)$   |
> > >
> > >
> > > **MLMP** belongs to the **Fully TTA setting**, which is both **more challenging** and **more realistic** for real-world deployment. It uses only unlabeled test images at inference, without labels, source data, or training-time changes, and remains effective even with a single test sample (Table 3 in the Appendix). Unlike FSS, which requires labeled support sets, MLMP is inherently capable of adapting on-the-fly to even changing domain shifts at inference time.
> > >
> > > We will include this comparison table and clarification in the final version of the paper. We hope this fully resolves the reviewer’s concern and more clearly highlights the distinct and practically relevant nature of the OVSS-TTA setting addressed in our work.
> > >
> > > [1] Tent: Fully Test-time Adaptation by Entropy Minimization, ICLR 2021

---

> > > > ### Comment · Reviewer_rgyA · 2025-08-05
> > > >
> > > > Thank's for the authors' response. This table solves my concerns about motivation and setting. Adding this table into the paper as a background introduction will make it more readable. I'll increase my score according to the authors' response.

---

### Official Review · Reviewer_t4bA · 2025-07-03

**Clarity:** 3
**Significance:** 3
**Originality:** 2
**Rating:** 4
**Confidence:** 4

**Summary:**

The work proposes a novel TTA method—Multi-Level and Multi-Prompt (MLMP) entropy minimization—that adapts VLMs for segmentation using intermediate vision-encoder features and diverse text prompts at both global and pixel levels. The proposed method is plug-and-play and requires no extra data or labels. It further  introduces a comprehensive OVSS TTA benchmark with 7 datasets, 15 corruptions, and 82 test cases. Experiments show that the proposed method consistently outperforms classification-based TTA baselines in this setting.

**Questions:**

My main concern is

1. how does the work select 7 prompts out of 80? Different selection might impact the performance.
2. Comparison should be shown on realistic ACDC dataset.
3. Technical novelty should be clarified.

**Ethical Concerns:**

["NO or VERY MINOR ethics concerns only"]

**Final Justification:**

Thanks for the nice work! I am recommending the final score of Broderline accept. Although the overall technical novelty appears limited agreed by other reviewers too, the task of test-time OVSS itself seems new, that might be relevant to the community. It further has detailed benchmark and analysis supporting the claims making it useful.

**Limitations:**

yes

**Quality:**

3

**Strengths And Weaknesses:**

Strength -

1. Exhaustive ablations and comparison: The work showcases exhaustive ablation and comparison showing that the proposed method works well.

2. It proposes new benchmark - test time adaptation for Open vocabulary segmentation model.

3. The work is nicely written and well motivated.


Weaknesses -

[A] Novelty: The technical contribution comprises of MLMP - that performs uncertainity aware entropy minimization of prediction probability across vision features, optimizing normalisation layers statistics. This has been the core idea of TENT work. Further, using multiple prompts[eg. OVSeg[1] have been used quite well for open-vocabulary segmentation. The technical novelty seems a bit limited.

[1] Open-Vocabulary Semantic Segmentation with Mask-adapted CLIP, CVPR 23

[B] Comparison dataset - Although the comparison is quite exhaustive, but mostly focuses on synthetic corruptions (gaussian noise, zoom, motion blur etc) and doesn’t focus much on real world datasets. Prior test time adaptation works [2,3,4] primarily show results on adaptation from Cityscapes -> ACDC [2,3], realistic but synthetic SHIFT[2]. Would be good to add results atleast on realistic ACDC dataset.

[2] Continual Test-Time Domain Adaptation, CVPR 22
[3] Distribution-Aware Continual Test-Time Adaptation for Semantic Segmentation, ICRA 24

[C] Selection of prompt: The work uses 7 prompt templates out of 80 CLIP standard prompts. However it doesnt seem straightforward to select the best performing set. How was this done? Further in these 7 prompts, there are prompts like - ‘a origami of <class_name>’, ‘a <class_name> in video-game’. How are such prompts helpful? They seem to out of domain for common corruption based images.

[D] Minor: Any reason why last layer does not have higher confidence for COCOStuff dataset, as such is seen for PASCAL and Cityscapes dataset (Fig 3). Generally one would assume, last layer features to have more weightage in segmentation task as shown for PASCAL and Cityscapes datasets.

---

> ### Author Rebuttal · Authors · 2025-07-31
>
> We thank the reviewer for their thoughtful evaluation and valuable feedback. We are pleased that they found the work to be well motivated and appreciated both the introduction of our OVSS TTA benchmark and the thorough experimental analysis. In what follows, we address the reviewer’s comments point by point.
>
> **A1. Clarifications on Novelty**
>
> We appreciate the reviewer’s comment and welcome the opportunity to clarify our technical contributions. We agree that TENT is a well-established approach for test-time adaptation in classification, and we do not consider entropy minimization itself to be a novel contribution, nor is it the core of our method.
>
> Our main contribution lies in **test-time adaptation of open-vocabulary vision-language models**, a fundamentally different and largely underexplored setting compared to standard classification. Due to the dense, spatial nature of segmentation and the prompt-sensitive behavior of VLMs, directly applying methods like TENT, WATT, or CLIPArTT is non-trivial and often ineffective. We address these challenges by proposing a set of novel and carefully designed components: (1) A **multi-layer adaptation strategy** that integrates features from intermediate vision-encoder layers to capture complementary, shift-resilient cues (2) An **adaptive uncertainty-aware weighting strategy** that dynamically upweights more confident layers and downweights noisier ones, allowing the model to emphasize the most reliable feature representations during test time (3) An **intuitive multi-prompt aggregation mechanism at the loss level** to address increased prompt sensitivity in OVSS. Each of these components is validated through detailed ablations (see Table 1-3), and together they yield consistent, substantial gains across a wide range of scenarios. Our method is also **highly efficient**, adding only marginal overhead compared to TENT (please see Appendix E).
>
> In addition to our design choices, we evaluate MLMP and adapted baseline methods exhaustively on a **comprehensive OVSS TTA benchmark suite** that now includes not only clean and synthetically corrupted data, but also realistic domain shifts (please see the t4bA-A3). This suite ensures fair and consistent evaluation, and we hope it will serve as a foundation for future TTA research in this domain.
>
> In summary, we believe that the novel design of MLMP, its **plug-and-play** and **efficient formulation**, and its **strong empirical gains** across diverse and realistic settings represent a meaningful step forward in the emerging and largely underexplored field of test-time adaptation for open-vocabulary segmentation. Moreover, MLMP is **modular** and **can be seamlessly applied to a wide range of contrastive VLM-based OVSS methods**, enabling lightweight and effective adaptation without requiring additional supervision. It proves effective not only under synthetic corruptions and real-world domain shifts, but also enhances performance on clean, natural images.
>
> **A2. Differences with Multiprompt used in OVSeg**
>
> We thank the reviewer for this helpful observation and take the opportunity to clarify key differences between our multi-prompt design and prior approaches such as OVSeg [1]. While OVSeg employs prompt ensembling, it does so during training by simply averaging text embeddings at the text encoder output, collapsing multiple linguistic features into a single representation. This results in a fixed perspective and lacks the diversity necessary for robust adaptation in a TTA setting.
>
> In contrast, our method operates entirely at test time, and for the first time, introduces a **loss-level integration of multiple prompts during test time adaptation**. We demonstrate that prompt variability induces significantly greater uncertainty in segmentation (Fig. 1a), especially at the patch and intermediate-layer levels. To exploit this signal effectively, we directly incorporate prompt diversity into our optimization objective, treating each template as an independent “critic” and averaging their losses. Using multiple prompts in this way acts as a form of cross-modal regularization, encouraging more stable and generalized learning signals. While different from traditional image augmentation, it can be viewed as a lightweight and safe augmentation technique in the text modality. This design is further supported by Proposition 1, which shows that the resulting gradient is both unbiased and lower-variance, thereby promoting more stable and generalizable adaptation.
>
> We also compare our approach in Table 3 to alternatives, including simple prompt averaging (as in OVSeg [1]) and parameter ensembling (as in WATT [2]), and show that our strategy outperforms both in adaptation effectiveness. While WATT incurs notable computational overhead, our method remains lightweight—even when using multiple templates—making it well-suited for dense prediction tasks such as OVSS.
>
> [1] Open-Vocabulary Semantic Segmentation with Mask-adapted CLIP, CVPR 23
>
> [2] WATT: Weight Average Test-Time Adaptation of CLIP, NeurIPS 24
>
> **A3. Including Real Domain Shift**
>
> We appreciate the reviewer’s suggestion to include evaluation on realistic, non-synthetic domain shifts. In response, we have conducted additional experiments on the ACDC [3] dataset, which contains real-world variations such as fog, night, rain, and snow. As in our main experiments, we perform test-time adaptation separately on both reference (clean) and shifted images within each domain. It is worth mentioning that in ACDC, half of the adverse-condition images have corresponding clean reference views. While these reference images are captured at approximately the same location as the shifted ones, the scene content may differ (e.g., different vehicles or objects present) due to real-world variability.
>
> As shown in the table below, our method yields consistent improvements over both baselines across all domains. Compared to the non-adapted model, MLMP achieves substantial gains, around 6% mIoU improvement on reference images and over 7% mIoU in the presence of environmental domain shift. These results highlight **the robustness of our method in handling real-world domain shifts, demonstrating effectiveness beyond synthetic corruption scenarios**. We also note that the Cityscapes dataset, on which we already reported results in the main paper, exhibits natural distributional shifts due to variations in lighting, camera viewpoints, and urban layouts across different cities.
> | Condition| No Adapt.| TENT| **MLMP**|
> |-|-|-|-|
> | Fog (ref)| 24.80 ± 0.00| 26.52 ± 0.03 | **31.28 ± 0.03** |
> | Fog| 23.88 ± 0.00| 26.89 ± 0.04 | **33.33 ± 0.04** |
> | Night (ref) | 24.95 ± 0.00| 26.54 ± 0.04 | **28.81 ± 0.10** |
> | Night| 22.12 ± 0.00| 24.17 ± 0.00 | **24.76 ± 0.03** |
> | Rain (ref)| 24.79 ± 0.00| 26.62 ± 0.00 | **31.10 ± 0.03** |
> | Rain | 23.86 ± 0.00| 26.84 ± 0.04 | **32.44 ± 0.04** |
> | Snow (ref)| 22.10 ± 0.00| 24.27 ± 0.04 | **28.22 ± 0.02** |
> | Snow | 23.54 ± 0.00| 27.25 ± 0.05 | **30.59 ± 0.03** |
> | **Avg (all ref)** | 24.16 | 25.99| **29.85**|
> | **Avg (all domains)**| 23.35 | 26.29| **30.28**|
>
> To further address the reviewer’s concern, we also evaluated MLMP on a rendered domain shift, using the GTAV dataset [4]. GTAV consists of photorealistic urban scenes rendered from a game engine and exhibits a distribution shift distinct from real-world imagery seen during CLIP pretraining. As shown in the table below, MLMP achieves consistent improvements over both TENT and the non-adapted baseline. This complements our ACDC results and demonstrates **MLMP’s robustness across both real-world and rendered test-time domains**.
> |Method|NACLiP|TENT|MLMP|
> |-|-|-|-|
> |mIoU (%)|25.09|26.62 ± 0.01| **28.84 ± 0.02**|
>
> [3] ACDC: The Adverse Conditions Dataset, ICCV 21
>
> [4] Playing for Data: Ground Truth from Computer Games, ECCV 16
>
>
> **A4. Selection of Prompts**
>
> We thank the reviewer for this important question. As discussed in Appendix C, the seven prompt templates used in our method were selected directly from the original CLIP repository, without any tuning or dataset-specific filtering. These prompts are **general-purpose** and were originally proposed to offer **diverse textual views** across a wide range of classes—not tailored to specific datasets or domains. We intentionally adopted this fixed, predefined set to avoid any risk of target-data information leakage
>
> While we agree that prompt selection can influence performance, selecting or filtering prompts based on the target domain shift would violate the fundamental assumptions of test-time adaptation, where no access to shift-specific information is allowed. In fact, using domain-aligned prompts (e.g., referencing weather conditions) could introduce bias and unfairly benefit adaptation. Our goal was to evaluate MLMP under a fair and domain-agnostic setting.
>
> **A5. Insights into Layer-Wise Confidence Trends in COCO-Stuff**
>
> We thank the reviewer for this insightful observation. Indeed, as shown in Fig. 3, the final vision layer receives lower confidence in the COCOStuff dataset compared to Pascal and Cityscapes. We believe this is due to the inherent properties of COCOStuff, which includes 171 classes, many of which are amorphous “stuff” categories (e.g., sky, grass, wall) with weak spatial boundaries and higher semantic ambiguity. In such cases, earlier and intermediate layers—preserving finer texture and edge information—often yield more reliable predictions than the deeper, class-aligned representations of the final layer. A similar trend is observed in the COCOObject dataset as well (please see Fig. 1 in the Appendix H).
>
> This observation further supports the necessity of our proposed **uncertainty-aware multi-level fusion strategy**, which dynamically reweights intermediate features based on their entropy. Unlike fixed fusion schemes, our approach adapts to the specific input and dataset, allowing the model to emphasize more reliable layers under varying conditions.

---

> > ### Comment · Reviewer_t4bA · 2025-08-01
> > **More explanation**
> >
> > Thank you so much for the rebuttal!
> >
> > I have further following doubts -
> >
> > 1. Selection of pronpt - Yes, totally agree about the general purpose prompts - but any reason why the work selects first 7? The result will be different if it selects first 10 or say first 5 or first 3.
> >
> > 2. COCO-Stuff layer wise confidence score - The rebuttal argues it is primarily due to many stuff classes like sky, grass, wall etc. However this seems counterintuitive - COCOObject does not have such classes and also have similar trends. Further Cityscapes have such classes but do not follow this trend.

---

> > > ### Author Response · Authors · 2025-08-04
> > >
> > > **1. Selection of Prompts**
> > >
> > > Thank you for the insightful question. We realize this may not have been fully clear in the paper: **the 7 prompts we use are not selected by us**, but directly adopted from the CLIP authors. As mentioned in their public implementation repository, they performed sequential forward selection over the full set of 80 templates and stopped after identifying 7 that gave the best ensemble performance.
> > >
> > > As noted in their repository "clip/notebooks/Prompt_Engineering_for_ImageNet.ipynb":
> > >
> > > *“After the 80 templates were ‘locked’ for the paper, we ran sequential forward selection over the list of 80 templates. The search terminated after ensembling 7 templates and selected them in the order below.”*
> > >
> > >  They further note that:
> > >
> > > *“This subset performs a bit better than the full 80 ensemble reported in the paper, especially for the smaller models.”*
> > >
> > > We follow this standard setup for consistency with prior work. In Figure 3, we also show that using the first 3 prompts from this set leads to lower performance, and that increasing the number beyond 7 can actually hurt results, suggesting that this particular selection strikes a good balance.
> > >
> > > ---
> > >
> > > **2. COCO-Stuff layer wise confidence score**
> > >
> > > We appreciate the reviewer’s thoughtful follow-up. To clarify, our intention was not to attribute the observed behavior solely to the presence of “stuff” classes in COCO-Stuff. They were cited only as a factor to contrast with datasets like Cityscapes. More specifically, we believe the main reason is that both COCO-Stuff (171 classes) and COCO-Object (80 classes) include a large number of categories, **many of which are really small and visually similar** (e.g., spoon, fork, knife).
> > >
> > > These characteristics make it more difficult for the final vision layers, optimized for high-level semantic abstraction, to produce confident predictions. In contrast, intermediate layers capture more local and fine-grained visual cues (e.g., texture, edges), which are more useful in such complex settings.
> > >
> > > On the other hand, datasets like Cityscapes have only 19 classes, most of which are **larger, spatially separated, and visually distinct** (e.g., car, person). Moreover, it features highly structured and well defined urban scenes, captured at high resolution. These properties make the final layers more reliable in this case.
> > >
> > > This difference in behavior is exactly what our method is designed to handle: in COCO datasets, our model automatically gives more weight to intermediate layers, while in datasets like Cityscapes, it relies more on the final layers, as shown in Figure 3. This adaptive weighting is central to the effectiveness of our multi-level design.

---

> ### Comment · Reviewer_t4bA · 2025-08-05
>
> Thanks for detailed rebuttal and responses. My concerns are addressed and I have raised my score to 4. While the overall technical novelty seems limited, the task of test time OVSS in general looks new with detailed benchmark and analysis.

---

> > ### Author Response · Authors · 2025-08-06
> >
> > Thank you for your thoughtful response, insightful suggestions and for engaging with our rebuttal. We’re glad your concerns were addressed and that you found our paper  well-motivated for the novel and underexplored task of OVSS-TTA, along with your recognition of the new, thorough benchmark we introduced and the exhaustive evaluation we conducted. We also appreciate your suggestion to include ACDC. We believe it is a valuable addition and will include it in the final version. We noticed that your message mentions raising the score to 4, which was already your initial rating, we just wanted to flag this in case it was a typo. In any case, we remain happy to address any further questions or feedback you may have.

---

### Note · Authors · 2025-08-15

We sincerely thank the AC and all reviewers for their time, effort, and constructive feedback throughout the process.

During the discussion, we made every effort to address all raised points. For **R-t4bA, R-rgyA, and R-Fuid**, we are pleased that our rebuttal addressed all their concerns, and for **R-tiCb**, we addressed most points in detail. Regarding the remaining concern of R-tiCb on the novelty of two specific components, multi-layer feature fusion and image-level entropy loss, we respectfully note that no prior work in segmentation has applied these design choices in an unsupervised TTA setting. We believe that introducing these simple yet powerful designs into our OVSS-TTA setting is itself a novel contribution. Additionally, we would like to highlight that **these components are not the core contributions of our work**. As noted by R-Fuid, **the method is recognized as both novel and well-motivated**. More specifically, our primary novelties are:
- An **uncertainty-aware fusion strategy that adaptively integrates intermediate layers at test time**, dynamically weighting them per input without labels, the first such approach in TTA, validated through detailed ablations.
- A **multi-prompt optimization applied directly in the adaptation loss**, explicitly leveraging prompt diversity during optimization, also a first in this setting, supported both by theoretical motivation and detailed ablations.
- A **comprehensive evaluation benchmark** that not only incorporates common corruptions but also includes realistic domain shifts, as suggested by R-t4bA. We believe this benchmark, along with the exhaustive evaluation presented in our paper, serve as a standardized testbed for future OVSS-TTA research and we respectfully consider this another important and novel contribution of our work.

We are happy that the reviewers found the paper **clear, well-written,** and **well-motivated**, and recognized our **novel evaluation benchmark for OVSS-TTA**, an area that, while highly practical, is largely unexplored. They highlighted the **interesting observations, vivid visualizations, solid theoretical analysis**, and **extensive experiments/ablations,** along with **consistent, substantial gains over baselines** across datasets, domain shift types, model sizes, and families.

Once again, we thank the reviewers for their constructive feedback and will incorporate all the valuable insights gained through the discussion into the final version of the paper.

---

### Decision · Program_Chairs · 2025-09-17

**Decision:**

Accept (poster)

**Comment:**

Paper was reviewed by four expert reviewers and received al positive scores of 3 x Borderline Accept and 1 x Accept. Initial reviews had a number of concerns that focused on (1) novelty, (2) lacking motivation, (3) lack of details (e.g., on how prompts are selected), (4) lack of comparisons to traditional methods, and (5) lack of clarity in certain places. However, rebuttal has largely addressed these concerns and reviewers appear to be satisfied with the provided responses.

AC has carefully considered the reviews, rebuttal, discussion and the paper itself. AC agrees with [t4bA] that overall technical novelty of the approach is limited. At the same time, the simplicity of the approach and the comprehensive benchmarking and evaluation are commendable. On balance AC is in agreement with reviewers and is recommending Acceptance.

Authors are strongly encouraged to incorporate content from the rebuttal, especially clarification and additional experiments, into the main paper (and supplementals where appropriate). This will strengthen the paper.